# Genetic diversity, antifungal evaluation and molecular docking studies of Cu-chitosan nanoparticles as prospective stem rust inhibitor candidates among some Egyptian wheat genotypes

Hanaa S. Omar[1,2]*, Abdullah Al Mutery[3,4], Neama H. Osman[1], Nour El-Houda A. Reyad[5], Mohamed A. Abou-Zeid[6]

**1** Faculty of Agriculture, Genetics Department, Cairo University, Giza, Egypt, **2** GMO lab Faculty of Agriculture, Cairo University, Research Park, CURP, Giza, Egypt, **3** Department of Applied Biology, College of Sciences, University of Sharjah, Sharjah, United Arab Emirates, **4** Molecular Genetics and Stem Cell Research Laboratory, University of Sharjah, Sharjah, United Arab Emirates, **5** Plant Pathology Department, Faculty of Agriculture, Cairo University, Giza, Egypt, **6** Wheat Disease Research Department, Plant Pathology Research Institute, ARC, Giza, Egypt

* Hanaa8324@yahoo.com

## Abstract

Wheat has a remarkable importance among cereals worldwide. Wheat stem and leaf rust constitute the main threats that destructively influence grain quality and yield production. Pursuing resistant cultivars and developing new genotypes including resistance genes is believed to be the most effective tool to overcome these challenges. This study is the first to use molecular markers to evaluate the genetic diversity of eighteen Egyptian wheat genotypes. Moreover, the molecular docking analysis was also used to assess the Cu-chitosan nanoparticle (CuChNp) treatment and its mode of action in disease control management. The tested genotypes were categorized into two main cluster groups depending on the similarity matrix, i.e the most resistant and susceptible genotypes to stem and leaf rust races. The results of SCoT primers revealed 140 polymorphic and 5 monomorphic bands with 97% polymorphism. While 121 polymorphic and 74 monomorphic bands were scored for SRAP primers (99% polymorphism). The genotypes Sakha 94, Sakha 95, Beni Sweif 4, Beni Sweif 7, Sohag 4 and Sohag 5 were resistant, while Giza 160 was highly susceptible to all stem rust races at the seedling stage. However, in the adult stage, the 18 genotypes were evaluated for stem and leaf rust-resistant in two different locations, i.e. Giza and Sids. In this investigation, for the first time, the activity of CuChNp was studied and shown to have the potential to inhibit stem and leaf rust in studied Egyptian wheat genotypes. The Spraying Cu-chitosan nanoparticles showed that the incubation and latent periods were increased in treated plants of the tested genotypes. Molecular modeling revealed their activity against the stem and leaf rust development. The SRAP and SCoT markers were highly useful tools for the classification of the tested wheat genotypes, although they displayed high similarities at the morphological stage. However, Cu-chitosan nanoparticles have a critical and effective role in stem and leaf rust disease control.

**Data Availability Statement:** All relevant data are within the manuscript and its Supporting Information files.

**Funding:** Our authors received no specific funding for this work.

**Competing interests:** NO authors have competing interests.

# 1. Introduction

Wheat (*Triticum aestivum*) is considered to be a major source of food grains in Egypt and worldwide. By 2050, wheat demand is expected to noticeably rise by 60 percent, particularly in the developing countries [1]. This increased demand for wheat is very serious, especially in the context of climate change, resulting in a 29% decrease in final productivity [2]. Furthermore, wheat is mainly targeted by three rust diseases, i.e. stripe, leaf, and stem rusts. The latter was caused by *Puccinia graminis* Pers. f. sp. *tritici* is a devastating disease in wheat crops around the world where it has become has epidemic under suitable environmental conditions [3]. Stem rust fungus attacks wheat plants, especially those planted lately, leading to blockage of the vascular system, plant stunting, and finally causing up to approximately 100% yield losses due to damaged grains and tillers [4]. In Egypt, this pathogen causes 1.96 to 8.21% losses in yield of most of the local wheat varieties [5]. In Egypt, leaf rust caused by *Puccinia triticina* could cause yield loss of 5–10% or more in commercial wheat cultivars as they lack adequate resistance to the pathogen [6,7].

Molecular marker techniques are considered a useful tool for the determination, characterization, and assessment of genetic diversity among different organisms at the species-level (Hanaa, *et al.* 2020). SCoT and SRAP markers are effectively used in various species of microorganisms to test the genetic diversity, structure, identification of cultivars, quantitative trait loci (QTL) mapping and DNA fingerprinting [8,9]. However, the genetic diversity assessment between and within the species of Carthamus was carried out by SRAP [10,11].

Bioinformatics and molecular docking approaches are important to identify new sources of resistant genes and to decipher the role of natural compounds such as chitosan-copper nanoparticles as natural fungicides that could serve as an environmentally friendly alternative to synthetic fungicides. Recent studies [12] have shown that a single MAP kinase. Pmk1 of *M. oryzae* has a chemical genetic inhibition, preventing the invasion of adjacent plant cells, and hence leaving the casual fungus trapped within a cell. Pmk1 regulates the expression of the secreted fungal effector proteins involved in host immune defense suppression, preventing reactive oxygen species production and excessive callose deposition at plasmodesmata. Moreover, it also regulates the hyphal constriction necessary for fungal growth from one cell to the adjacent rice cell, facilitating the colonization of the host tissue and blast disease. Besides, MAPK could also be used for the production of a natural component as the effective fungicides against wheat leaf rust pathogen as a molecular target.

Chitosan mainly works on plant innate immunity that makes it resistant to most pathogen and plant diseases. This property could additionally be improved by utilizing it as nanoparticles. Moreover, chitosan can interact with several metals, establishing different chemical bonds that increase its stability when converted to nanoparticles. Metal-based chitosan nanomaterial plays a dual role as a plant growth promoter and a plant protection agent as well. Due to their dual activity, metal-based chitosan nanomaterial have attracted a lot of interest. Chitosan encapsulated metals are also less toxic due to the process of slow-release and have also shown a long-lasting impact on plants. Mixing Zn and Cu with Nano-chitosan is beneficial as it improves plant nutrition and growth, and protects against both biotic and abiotic stresses [13]. In this connection, the application of copper-chitosan nanoparticles could protect tomato plants from blight and wilt pathogens [14]. However, the effect of CuChNp on controlling each leaf and stem rust of wheat has not been identified yet. Therefore, the main objectives of this study are to evaluate the disease response of 18 Egyptian wheat varieties against the bioagent stem and leaf rust at the adult plant stage, under field conditions and at the seedling stage under greenhouse conditions. Moreover, to assess the genetic diversity among eighteen species of wheat plants using both SRAP and SCoT molecular markers. Also, to determine the

relationship between the disease parameters and molecular markers linked to them. Additionally, a preliminary screening and evaluation process of the copper-chitosan nanoparticles antifungal on 18 Egyptian wheat varieties against stem rust was performed and their mode of action was studied by molecular docking analysis. Therefore, further studies are needed to completely and crucially emphasize and investigate the protective role of these compounds in controlling such a disease in wheat.

## 2. Materials and methods

Ethics statement not applicable in this the present study.

### 2.1 Assessment of the disease reaction of the tested wheat genotypes

**2.1.1 Assessment of the reaction of wheat seedlings to the stem and leaf rust diseases (greenhouse experiment).** This experiment was carried out at the seedling stage in the greenhouse of Wheat Diseases Res. Dept., Plant Pathology Res. Inst. and ARC Giza, Egypt during the year 2020.

Eighteen Egyptian wheat genotypes were evaluated for both stem and leaf rust resistance, as shown in Table 1. In three complete random replicates, these genotypes were screened against twenty stem rust and leaf rust races. Six grains of each wheat genotype were sown in plastic pots (6 cm. diam.) containing a mixture of peat moss and soil (1:1 v/v). A paintbrush infected with the urediniospores of each rust race inoculated the seven-day-old seedlings. The inoculated seedlings were incubated for 14 hrs in a dark dew chamber at 18° C, then transferred to the greenhouse benches and maintained at 20–24° C and 70–80 percent relative humidity with 16 hrs of light and 8 hrs of dark at around 7600 lux [15]. Sowing was observed regularly before rust and leaf pustules formed. The seedlings' response was scored 10–12 days after inoculation based on the type of infection (IT) expressed using a 0–4 scale on each genotype [16].

**Table 1. Pedigree and year of release of the tested wheat genotypes.**

| Code | Wheat genotype | Pedigree | Year of release |
|------|----------------|----------|-----------------|
| 1 | Gemmeiza 11 | B0W"S"/KVZ"S"//7C/SERI82/3/GIZA168/SAKHA61. GM7892-2GM-1GM-2GM-1GM-0GM. | 2011 |
| 2 | Gemmeiza 12 | OTUS/3/SARA/THB//VEE.CCMSS97Y00227S-5Y-010M-010Y-010M-2Y-1M-0Y-0GM | 2017 |
| 3 | Sids 12 | BUC//7C/ALD/5/MAYA74/ON//1160-147/3/BB/GLL/4/CHAT"S"/6/MAYA/VUL-4SD-1SD-1SD-0SD. | 2007 |
| 4 | Misr 1 | OASIS/SKAUZ//4*BCN/3/2*PASTOR. CMSSOYO1881T-050M-030Y-O3OM-030WGY-33M-0Y-0S. | 2010 |
| 5 | Misr 2 | SKAUZ/BAV92. CMSS96M0361S-1M-010SY-010M-010SY-8M-0Y-0S. | 2011 |
| 6 | Misr 3 | ATTILA*2/ABW65*2/KACHU CMSS06Y00258 2T-099TOPM-099Y-099ZTM-099Y-099M-10WGY-0B-0EGY | 2018 |
| 7 | Giza 168 | MIL/BUC//Seri CM93046-8M-0Y-0M-2Y-0B | 1999 |
| 8 | Giza 171 | Sakha 93 / Gemmeiza 9 S.6-1GZ-4GZ-1GZ-2GZ-0S | 2013 |
| 9 | Sakha 94 | OPATA/RAYON//KAUZ. CMBW90Y3280-0TOPM-3Y-010M-010M-010Y-10M-015Y-0Y-0AP-0S. | 2004 |
| 10 | Sakha 95 | PASTOR//SITE/MO/3/CHEN/AEGILOPS SQUARROSA(TAUS)//BCN /4/WBLL1CMSA01Y00158S-040P0Y-040M-030ZTM-040SY-26M-0Y-0SY-0S | 2018 |
| 11 | Beni Sweif 7 | ALO/5/HUI/YAV_1/6/ARDENTE/7/HUI/YAV79/8/POD_9CDSS02Y01233T0OTOPB-0Y-0M-26Y-0Y-0SD | 2017 |
| 12 | Shandaweel 1 | SITE//MO/4/NAC/TH.AC//3*PVN/3/MIRLO/BUC. CMSS93B00567S-72Y-010M-010Y-010M-0HTY-0SH | 2011 |
| 13 | Giza 164 | KVZ/Buha "s"//Kal/Bb CM33027-F-15M-500y-0M | 1987 |
| 14 | Sakha 69 | Inia–RL4220´7C/YR"S"CM15430- 25 -65-0S-0S | 1980 |
| 15 | Giza 160 | Chenab 70/Giza 155 | 1982 |
| 16 | Beni Sweif 4 | AUSL/5/CANDO/4/BY*2/TACE//II27655/3/TME//ZB/W*2. ICD88-1120-ABL-0TR-1BR-0TR-6AP-OSD. | 2007 |
| 17 | Sohag 4 | Ajaia-16//Hora/Jro/3/Gan/4/Zar/5/Suok-7/6/Stot//Altar84/Ald CDSS99B00778S-OTOPY-0M-0Y-129Y-0M-0Y-1B-0SH | 2016 |
| 18 | Sohag 5 | TRN//21563/AA/3/BD2080/4/BD2339/5/Rascon 37// Tarro 2// Rascon 3/6/Auk/Gull//GreenCDSS00B00364T-0T0PB-0B-2Y-0M-0Y-1B-0Y-0SH | 2016 |

**2.1.2 Assessment of adult plant resistance to wheat stem and leaf rust diseases (field experiment).** This study was performed at Giza and Sids, Agricultural Research Stations, Agric. Res. Center of Egypt, during the 2020 growing season in a randomized complete block design (RCB), with three replicates. Each genotype was sown in plots divided into two rows of a three-meter length and 30 cm apart. All plots were surrounded by a highly susceptible wheat variety, i.e. Morocco and Thatcher plants, which were planted over an area of 1 m$^2$ to provide a high and uniform disease pressure. All recommended cultural practices, i.e. fertilization, irrigation and other management were applied. To enhance the development of stem rust epidemics under field conditions, the experiment was irrigated and inoculated by dusting a mixture of urediniospores of the common and more aggressive stem rust pathotypes, mixed with talcum, at a rate of one volume of fresh urediniospore mixture to 20 volumes of talcum powder according to the method described by Tervet and Cassell [17]. The development of the disease was assessed using three epidemiological parameters, i.e. final rust severity (FRS %), area under disease progression curve (AUDPC) and stem and leaf rust disease increasing rate increase (R-value). The adult-plant reactions were reported using the [18] description. When the highly susceptible (check) variety was severely rusted and the disease rate reached its maximum or final level of severity, the final rust severity (FRS %) was recorded for each of the tested varieties [19]. The area under the disease progress curve (AUDPC) disease was estimated using the following formula proposed by Pandey *et al* [20]. In addition, the rate of increase in stem and leaf rust diseases (R-value) was determined using the following [21].

AUDPC = D [1/2 (Y1 + Yk) + (Y2 + Y3+ - - - - - + Yk-1)]

Where: D = Days between two consecutive recordings (time intervals)

Y1 + Yk = Sum of the first and last disease severity scores.Y2 + Y3 + - - - + Yk-1 = Sum of all in between disease scores.

$$\text{r-value} = \frac{1}{t_2 - t_1} \left( \log_e \frac{X_2}{1 - X_2} - \log_e \frac{X_1}{1 - X_1} \right)$$

X1 = the proportion of the susceptible infected tissues (disease severity %) at date $t_1$.
X2 = the proportion of the susceptible infected tissues (disease severity %) at date $t_2$, $t_2$-$t_1$
T1 = the interval in days between these dates.

## 2.2 Molecular analysis

**2.2.1 DNA extraction.** The high-quality genomic DNA was isolated from the fresh eighteen wheat leaf genotypes (100 mg) using the CTAB method [22]. Spectrophotometer analysis was used to measure the DNA concentrations (260 / 280). The gel electrophoresis (1% agarose gel) was used for PCR analysis at the final 25 ng/μl concentration. A 100 bp DNA ladder was used as a DNA marker.

**2.2.2. SCoT -PCR amplification.** A total of 20 primers established by Collard and Mackill [23] were utilized for genetic diversity analysis of the 18 wheat genotypes. The PCR technique was done in a total volume of 25 μl containing 1X reaction buffer, 1.5 mM MgCl2, 0.1 mM dNTP, 0.3 μM of a primer, 60 ng genomic DNA and 2U of Taq DNA polymerase. The amplification of PCR was programmed at 94 °C for 3 min, 36 cycles of 94 °C for 50 °C for 1 min and 72 °C for 2 min and the final step at 72 °C was held for 5 min. All the PCR amplification products were separated by electrophoresis on 1.5% agarose gels.

**2.2.3. SRAP-PCR amplification.** The SRAP molecular markers analysis was performed according to the technique described by Li and Quiros [24] with minor modifications. All SRAP primer combinations were initially screened using a group of samples according to [25].

Twelve primer combinations with scorable polymorphic bands were done. PCR analysis was done in a total volume of 25 μl i.e. 10 μl of Master Mix, 2 μl of genomic DNA, 2 μl of the two set primers (10 μmol/l primers), and 3 μl of dd H2O. The SRAP markers were identified at the following parameters: 5 min at 94˚C, 5 cycles of 94˚C for 1 min, 35˚C for 1 min, 72˚C for 2 min, 30 cycles of 94˚C for 1 min, 50˚C for 1 min, 72˚C for 2 min, and final step of 5 min at 72˚C. The amplification fragments were separated by electrophoresis in a 1.5% agarose gel.

**2.2.4 Statistical analysis of data.** The PCR products produced from SCoT and SRAP analysis were recorded. The data analysis and the final scores were determined for the clear bands only. For each marker, the band was scored (1) as a present or (0) as absent to generate the binary data set for the eighteen wheat genotypes. The percentage of polymorphism was calculated by dividing the number of polymorphic fragments by the total number of amplified bands using the same primer or the primer combination. The genetic similarity coefficient was measured according to the Dice coefficient [26]. A dendrogram was produced through the cluster analysis using the unweighted pair group analysis of the mathematical average (UPGMA) for all the exclusive marker systems. systat ver. 7 (SPSS Inc. 1997 SPSS Inc.3/97 standard version statistical software analysis was used to compare the similarity matrices and the dendrograms generated by the SCoT and SRAP analysis for the 18 wheat varieties.

In addition, the significant differences between the eighteen varieties were verified by the analysis of variance (ANOVA) test as outlined by Snedecor and Cochran [27]. Mean comparisons for variables were made between the tested genotypes using the least significant differences (LSD at 5% level) value. On the other hand, based on the interaction between the presence of rust strains and wheat varieties, were developed. The heatmap was drawn with the aid of the ClustVis tool [28,29] and JavaScript script language [30].

## 2.3 Effect of nanoparticle composite on wheat stem rust

**2.3.1 Preparation of Cu–chitosan composite nanoparticle infection.** Cu–chitosan NPs were prepared based on the ionic gelation of chitosan with TPP anions and CuSO4 [31,32]. In a nutshell, chitosan at 0.5% (w/v) was dissolved in an acetic acid solution of 1% (v/v) and the pH was adjusted to 4.6–4.8 with 10 N sodium hydroxide (NaOH). Tri-Poly-Phosphate (TPP) solution (0.25%, w/v) was prepared and gradually added to 3 ml of chitosan solution in drops under magnetic stirring at room temperature then chitosan nanoparticles formed spontaneously. Before purification of the chitosan nanoparticle suspension, 100 μg/ml of copper ions solution was added and the final formed solution was centrifuged at 9000 g for 30 min at 4˚C (discard the supernatants). Then, the chitosan nanoparticles were extensively rinsed with distilled water to remove any sodium hydroxide, followed by sonication to gain purified Cu-chitosan NPs. The nanoparticles were dried using freeze-drying and stored for further use. Synthesized NPs were characterized for physicochemical properties, including particle size, using Transmission Electron Microscopy (TEM).

**2.3.2 Characterization of nanoparticles using Transmission Electron Microscopy (TEM).** TEM micrographs were obtained using an FEI Spirit TEM (Hillsboro, USA) operated at 120kV using a 400-mesh Formvar ⓡ carbon-coated copper grid. The Cu–chitosan NPs sample was prepared by vortexing and placing 2.0l of the sonicated colloidal solution onto the grid, using a 10 ml disposable pipet to re-suspend the sample, making EM grids (carbon-coated 400-mesh copper grids) directly on the specimen and using the filter to wick away specimen droplets, specimen–side up in the specimen petri dish.

**2.3.3 Seedling treatment and disease measurement.** The studied 18 wheat genotypes were sprayed with Cu-chitosan composite nanoparticles at 0.1% (w/v) concentration [33], either 24 hr. before or 24 hr. before and after inoculation with urediniospores of stem rust to

determine the effect of this solution and its application in controlling the disease. The estimation of the causal pathogen races as expressed on the wheat plants of the tested wheat genotypes was carried out as incubation period (IP), latent period (LP), and infection type (IT).

## 2.4. *In silico* interpretation of the Cu-chitosan composite nanoparticles against *P. graminis* f. sp. *tritici*

**2.4.1. Retrieve and analyze the target protein's sequence.** The sequence of each fungus protein of PtMAPK1 of *P. triticina* (Accession No. AAY89655.1) and PgMAPK of *P. graminis* f. sp. *tritici* (Accession No. EFP88010) was downloaded from NCBI (https://www.ncbi.nlm.nih.gov/) in FASTA format. NCBI BLASTp (https://blast.ncbi.nlm.nih.gov/) was used to identify the required template structure for protein modeling and functional prediction [34]. BLASTp analysis suggested that 2FA2 and 2VKN are the most appropriate template structure for the construction of the target proteins with the highest sequence identity and query coverage and less E value.

**2.4.2. Homology modeling of *Pt* MAPK1 and PgMAPK.** In the absence of a crystallographic structure for MAPK1 and MAPK protein sequences from P. *triticina* and *P. graminis f. sp. Tritici*, respectively, its three dimensional structure was acquired by homology modeling, according to a previously reported procedure [35]. Concisely, PtMAPK1 and PgMAPK primary sequences (GenBank entry: AAY89655.1 and EFP88010, respectively) were submitted to Swiss [36]. The SWISSMODLE server (https://swissmodel.expasy.org/) was used to predict the 3D structure of the MAPK1 and PgMAPK proteins using its template structure. The fungus proteins with a QMEAN score were established for the development of models. In addition, for final confirmation, the protein model with a below -4 score was chosen. The predictive model of MAPK1 and PgMAPK was eventually validated using Ramachandran plot.

**2.4.3. Evaluation of stability and reliability of the structural model.** To assess their stability and reliability, a range of methods were used to estimate the efficiency of the rough MAPK model. PROCHECK analysis, which quantifies the residues in the Ramachandran plot accessible zones, was used to identify stereochemical quality of the model. The ERRAT tool, which defines the protein's overall quality factor, was used to study statistics on no-bonded interactions among different types of atoms.

**2.4.4. Preparation of protein and ligands.** The ligand structures of the cu-chitosan composite nanoparticle treatments were determined to identify the direct effect on the fungi. The ligands were retrieved from the PubChem database. The ligands were prepared and designed in ChemDraw (https://www.perkinelmer.com/category/chemdraw) in PDB format, then converted into MOL2 format using openable software (http://openbabel.org/wiki). Then the ligands were iteratively docked to the homology model of MAPK1 and MAPK protein sequences from P. triticina and P. graminis f. sp. tritici, respectively. The molecular docking analysis was performed using SAMSON software 2020 (https://www.samsonconnect.net/) to determine the interaction between the target proteins of the wheat stem rust fungus. The affinity minimization was done using the 3DREFINE server (http://sysbio.rnet.missouri.edu/3Drefine/index.html). The energy minimization was done at neutral pH 7.0 ± 2.0.in SAMSON software. The grid was always configured with the following parameters i.e. all water molecules and ligands were deleted while hydrogen atoms for the target proteins were added. A receptor grid X, Y, and Z values of proteins were produced depending on the blind docking to explore the effect of the ligands on the proteins and the mechanism against the virulence of the causal pathogen.

**2.4.5 Binding site prediction (Protein-ligand docking).** The SAMSON software (https://www.samson-connect.net/) was used for binding site prediction. It uses the interaction energy

between the protein and a simple Van der Waals probe to locate energetically favorable binding sites. This software uses Auto-docking Vina as an element to maximize the accuracy of these predictions while minimizing the computer time. The program works based on quantum mechanics. It predicts the potential affinity, molecular structure, geometry optimization of the structure, vibration frequencies of coordinates of atoms, bond length, and bond angle. Following the exhaustive search, 100 poses were analyzed and the best scoring poses were used to calculate the binding affinity of the ligands. The ligands that tightly bind to a target protein with a high score were selected. The proteins were docked against the compounds chitosan and cu using SAMSON software (https://www.samson-connect.net/). The 2D interaction was performed to detect the favorable binding geometries of the ligand with the proteins using Discovery studio software to generate the 2D interaction image of the Docked protein-ligand complex with a high score to the predicted active site. The ligands were docked with the target protein and the best docking poses were identified.

## 3. Results

### 3.1 Evaluation of plant diseases

**3.1.1. Assessment of seedling resistance against the wheat stem and leaf rust pathogens.** In this study, eighteen Egyptian wheat genotypes were evaluated in their seedling stage against twenty different stem and leaf rust races under greenhouse conditions in Giza locations during the two growing season 2020. At the seedling-plant stage, twenty stem rust disease data on infection types and severity was determined on eighteen Egyptian wheat genotypes as presented in Table 2. Ten genotypes were found resistant (Misr 3, Sakha 94, Sakha 95, Beni Sweif 4, Beni Sweif 7, Sohag 4, Sohag 5, Gemmeiza 12, Shandaweel 1 and Giza 171).), accounting for 55% of total genotypes, eight genotypes (44%) (Gemmeiza 11, Sids 12, Misr 1, Misr 2, Sakha 69, Giza 164, Giza 168 and Giza 160) were susceptible. However, Misr 3, Sakha 94, Sakha 95, Beni Sweif 4, Beni Sweif 7, Sohag 4 and Sohag 5 genotypes were resistant to all races, but Giza 160 proved to be susceptible to all races as presented in Table 2. On the other hand, at the seedling-plant stage, five leaf rust disease date on infection types and severity was also detected on eighteen Egyptian wheat genotypes as shown in Table 3. Seven genotypes (38%) were noticed resistant (Gemmeiza 12, Misr 1, Giza 171, Misr 3, Beni Sweif 7, Giza 160 and Ben Sweif 4), while, eleven genotypes (61%) (Gemmeiza 11, Sids 12, Giza 164, Misr 2, Giza 168, Sakha 94, Sakha 69 Sakha 95, Shandaweel 1, Sohag 4 and Sohag 5) were susceptible.

Moreover, using the heat map to determine the relationship between the eighteen wheat genotypes and twenty stem rust races were performed. The results of Heat map and Principal component (PCA) as analyses revealed that one-dimensional heatmap visualization of the interaction between the presence of rust races and wheat varieties performance revealed grouping of the 18 wheat varieties into three distinct groups as presented in Figs 1 and 2. Cluster I consist of almost resistance to stem rust in the seedling stage; cluster II included moderate susceptible to stem rust races in the seedling stage and cluster III included the highly susceptible to stem rust races in seedling wheat genotypes. In this respect, the results confirmed that there is a correlation between the genotypes and the stem rust at the seedling. The results were quite similar to those from the previous analysis, with the inclusion of the term "response to seedling stage, All the terms are related to the response of plants to stem rust races disease.

**3.1.2. Postulation of genes (*Sr᾽s*) conditioning stem rust resistance at seedling stage under greenhouse conditions.** The possible *Sr, s* genes that condition resistance to stem rust in the 18 Egyptian genotypes were predicted and determined during this study as presented in Table 4. To postulate sr genes in the tested genotypes, the low (L) and high (H) infection types displayed by the eighteen wheat genotypes as presented in Table 5 were compared with the

**Table 2. Response of 18 wheat varieties against twenty stem rust races at the seedling stage under greenhouse condition.**

| Wheat genotypes | BTSTC | SKTTC | IKGKC | SIDCC | DKTTC | PKTTC | PKTTH | PCTKC | STKTC | TKTPC | TKPTC | TKSTC | TKTTS | TTKSC | TTTSK | PKTTH | PCTKC | STKTC | TKTPC | TKPTC |
|---|---|---|---|---|---|---|---|---|---|---|---|---|---|---|---|---|---|---|---|---|
| Gemmeiza 11 | 0 | 3 | 0 | 0 | 1 | 1 | 4 | 3 | 0 | 0 | 0 | 4 | 0 | 3 | 3 | 0 | 4 | 0 | 3 | 0 |
| Gemmeiza 12 | 0 | 2 | 0 | 0 | 2 | 0 | 2 | 2 | 4 | 2 | 0 | 0 | ;0 | 0 | 2 | ;0 | 0 | ;0 | 0 | 0 |
| Sids 12 | 2+ | 3 | 3 | 0 | 3 | 4 | 3 | 4 | 2 | 0 | ;0 | 4 | 4 | 3 | 3 | 4 | 4 | 0 | 3 | 3 |
| Misr 1 | 1 | 3 | 3 | 1 | 3 | 0 | ;0 | 4 | 4 | 1 | 4 | 2 | 2 | 3 | 1 | 0 | 3 | 3 | 1 | 0 |
| Misr 2 | 1 | 4 | 3 | ;0 | 4 | 2+ | 1 | 3 | 3 | 0 | 3 | 1 | 1, 2 | 3 | 2 | ;0 | 3 | 3 | 0 | ;0 |
| Misr 3 | 0 | 2+ | 2 | 0 | 0 | 0 | 1 | 0 | 1 | 2++ | 0 | 1 | 1 | 1 | 1 | 0 | 0 | ;0 | 0 | 0 |
| Giza 164 | 3 | 1 | 3 | 0 | 1 | 0 | 4 | 4 | 3 | 0 | 4 | 3 | 4 | 3 | 0 | ;0 | 4 | 0 | 4 | ;0 |
| Giza 168 | 0 | 4 | 0 | 1 | 1 | 1 | 4 | 3 | 4 | 0 | 4 | 4 | 3 | 3 | 3 | 0 | 3 | 0 | 3 | 0 |
| Giza 171 | 0 | 2 | 2 | ;0 | 0 | 0 | 4 | 2 | 4 | 0 | 4 | 2 | 3 | 2 | 1 | 0 | 4 | ;0 | 0 | 0 |
| Sakha 69 | 0 | 1 | 3 | 1 | 0 | 2++ | 4 | 3 | 1 | 2 | 4 | 4 | 1 | 0 | 1 | 0 | 3 | 0 | 3 | 0 |
| Sakha 94 | 2 | 0 | 1 | 0 | 0 | 0 | 1 | 2 | 2 | 2 | 0 | 0 | ;0 | 0 | ;0 | ;0 | 0 | 0 | 0 | 0 |
| Sakha 95 | 0 | 0 | 1 | ;0 | 1 | 0 | 1 | 1, 2 | 1 | 1 | 0 | 0 | 0 | 0 | 1 | 0 | ;0 | ;0 | 0 | 0 |
| Shandaweel 1 | 2 | 2 | 0 | 2 | 4 | 2 | 1 | 1 | 1 | 1 | 1, 2 | 0 | 1 | 1 | 3 | 3 | 0 | 0 | 0 | 4 |
| Beni Sweif 4 | 0 | ;0 | 1 | ;0 | 1, 2 | 0 | 0 | 1 | 2 | ;0 | 1 | 1 | 1 | 2 | 1 | 0 | 0 | 0 | 0 | ;0 |
| Beni Sweif 7 | ;0 | 1 | 1 | 0 | 2 | 1 | 0 | ;0 | 1 | 1 | 1 | 0 | 0 | 2++ | 1 | ;0 | 0 | ;0 | 0 | 0 |
| Sohag 4 | 1 | 1 | 1 | ;0 | 2 | 2 | 2 | 2 | 2++ | 2 | 2 | 0 | 2++ | 2++ | 2 | 2 | 0 | 0 | 0 | 0 |
| Sohag 5 | 1 | 1 | 2 | 0 | 0 | 0 | 1 | 1 | 2 | 0 | 1 | 1 | 2 | 2 | 1 | 0 | 0 | 0 | 0 | 0 |
| Giza 160 | 3 | 3 | 3 | 4 | 3 | 3 | 4 | 3 | 3 | 3 | 3 | 3 | 4 | 3 | 3 | 4 | 3 | 3 | 3 | 4 |

**Table 3. Response of 18 wheat varieties against five leaf rust races at the seedling stage under greenhouse condition.**

|  | DTJHC | JKGTC | TJTPC | TKKTC | PKPTC |
|---|---|---|---|---|---|
| Gemmeiza 11 | 4 | 2 | 4 | 4 | 4 |
| Gemmeiza 12 | 1 | 0 | 1 | 1 | 2 |
| Sids 12 | 0 | 4 | 2 | 0 | 4 |
| Misr 1 | 0 | 1 | 0 | 0 | 0 |
| Misr 2 | 0 | 2 | 0 | 4 | 0 |
| Misr 3 | 1 | 0 | 0 | 2 | 1 |
| Giza 164 | 4 | 4 | 4 | 4 | 4 |
| Giza 168 | 4 | 4 | 4 | 4 | 4 |
| Giza 171 | 1 | 2 | 0 | 2 | 1 |
| Sakha 69 | 4 | 4 | 4 | 4 | 4 |
| Sakha 94 | 4 | 3 | 3 | 3 | 4 |
| Sakha 95 | 1 | 2 | 4 | 1 | 3 |
| Shandaweel 1 | 4 | 2+ | 3 | 4 | 4 |
| Beni Sweif 4 | 0 | 1 | 0 | 2++ | 0 |
| Beni Sweif 7 | 0 | 1 | 1 | 2+ | 0 |
| Sohag 4 | 0 | 0 | 0 | 1 | 0 |
| Sohag 5 | 1 | 0 | 0 | 1 | 1 |
| Giza 160 | 4 | 4 | 4 | 4 | 4 |

infection types of some known *Sr's* genes against twenty identified tested pathotypes of *P. triticina* under greenhouse conditions. The wheat genotype Gemmeiza-11 probably possesses *Sr's* 5, 9a, 26, 17, 32 and other genes that indicate this cultivar showed low infection types to 16 phenotypes. However, the wheat genotype Gemmeiza 12 was resistant to eight pathotypes. It probably has *Sr's* 11, 9g, 30, 28, 29, 33 and other genes for stem rust resistance. Although the Misr 1 genotype was resistant to three phenotypes, it did not carry any resistance genes of the tested Sr line set. Misr 2 was resistant to 7 phenotypes. Thus, it probably has two genes, i.e. *Sr*11 and 9, and may have an additional gene (s). Misr 3 showed low infection types of 16 phenotypes. Thus, it may probably carry *Sr's* 5, 9, 26, 17, 32 and other genes for stem rust resistance. Giza 164 was resistant to three phenotypes, but it did not carry any resistance genes of the tested *Sr's*. Giza 168 was resistant to eleven phenotypes. And hence, it probably has *Sr's* 5, 11, 9g, 26, 30, 17, 28, 29, 33 and other genes from stem rust resistance. Giza 171 may probably carry all genes under study, i.e. *Sr* 5, 11, 36, 9a, 11, 6, 8a, 9g, 26, 9b, 30, 17, 28, 29, 30, 32, 33, and 34. Sakha 69 was resistant to four phenotypes, but it probably carries only one resistance gene, (*Sr* 29) and there could be other genes for leaf rust resistance. Sakha 94 showed a low infection type for ten phenotypes of the thirteen phenotypes under study. Therefore, this cultivar probably carries seven genes, i.e. *Sr's* 5, 11, 9g, 30, 28, 29, 33 and other genes for leaf rust resistance. Sakha 95 was resistant to 11 phenotypes. So, it probably has eight genes, i.e. *Sr's* 5, 11, 9g, 26, 30, 28, 29, 33 and may have an additional gene (s). Shandawel-1was resistant to 7 phenotypes. So, it probably has two genes, i.e. Sr11 and 9g, and may have an additional gene (s). Bani Swif 4 may probably carry Sr 5 and other genes for stem rust resistance. Beni Suef 7 may probably carry *Sr's* 9g, 26, 17 and other genes for stem rust resistance. Sohag 4 may probably carry Sr 5 and other genes for stem rust resistance. Sohag 5 has 4 genes (s) of the tested *Sr's* lines set. It probably has *Sr's* 29, 30, 33, 34 and other genes for leaf rust resistance. Giza 160 was resistant to three pathotypes, but it did not carry any resistance genes of the tested *Sr's* lines set. The data presented in S1 and S2 Tables indicated that the virulence genes (*Lr*s and *Sr*s) of

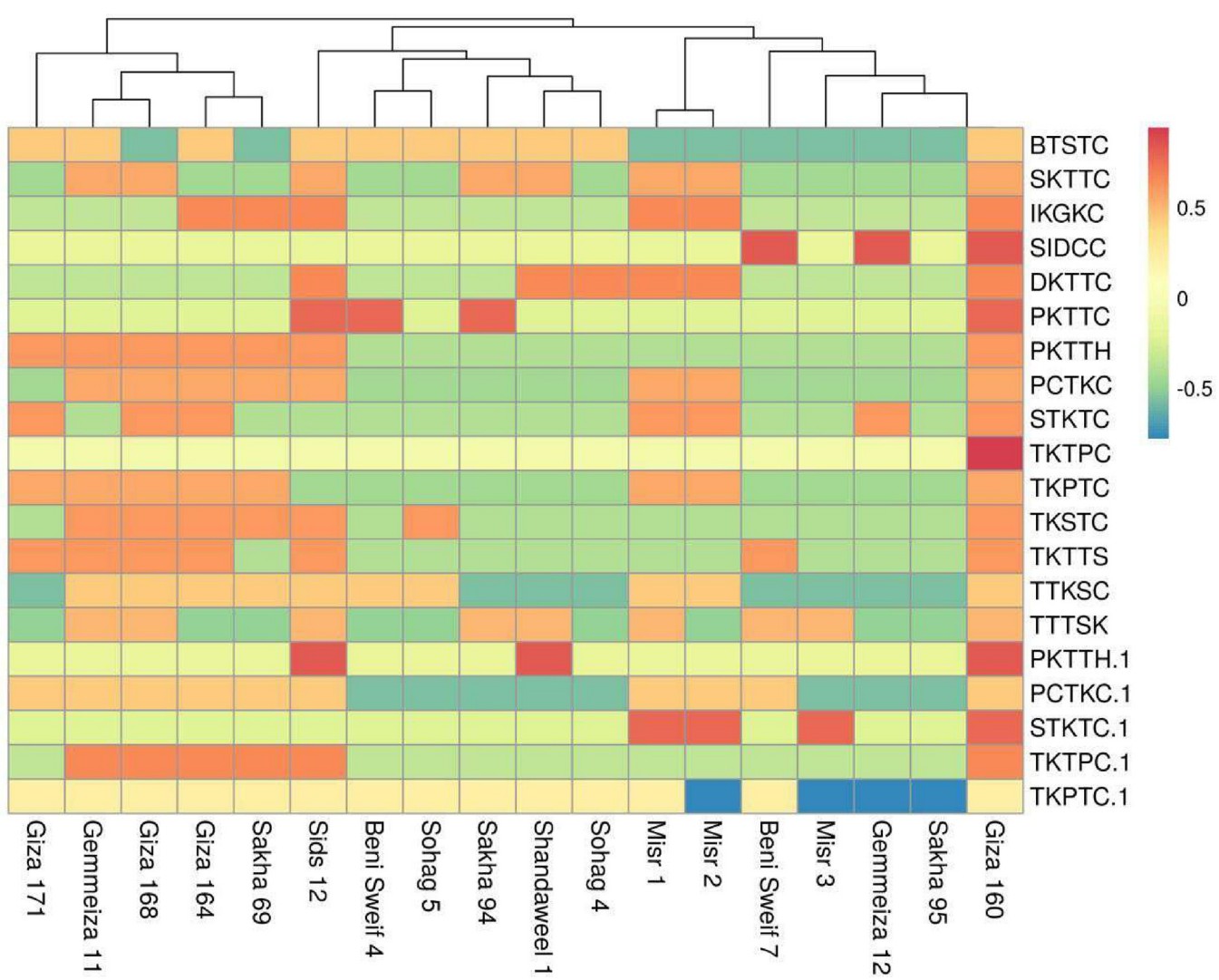

**Fig 1. One-dimensional heat map showing the clustering of the 18 wheat varieties based on the presence of rust strains and varieties' performance revealed grouping into three distinct groups.** Rows represent the 18 wheat genotypes and columns represent the 20 rust strains.

the 20 *P. graminis* f. sp. *tritici* and five *P. triticina* pathotypes were used to study stem and leaf rust resistance of eighteen Egyptian wheat genotypes. The wheat genotypes Gemmeiza-11, Gemmeiza-12, Giza 171, Sakha-94, Sakha 95, Sids 12, Misr 3, Beni Sweif 7, Beni Sweif 4, Sohag 4 and Sohag 5) probably possess more than three *Sr's* genes and other genes. And hence, it probably has more than *Sr's* other genes for stem rust resistance. That indicated this cultivar showed low infection types and resistance to many phenotypes. The result of gene Postulation confirmed the assessment of seedling resistance against the wheat stem rust pathogens.

**3.1.3 Assessment of wheat adult plant resistance to stem and leaf rust diseases under field condition.** The 18 wheat genotypes were evaluated for stem rust and leaf rust resistance at the adult plant stage under natural infection in the field at two different locations (Giza and Sids). The evaluation was carried out and expressed through these final rust severity (FRS %), the area under disease progress curve (AUDPC) and the rate of rust disease increase (R-value) as presented in Table 6. The results obtained from the final rust severity (FRS %) represented in Table 6 revealed in general that the FRS% was significantly varied among the tested

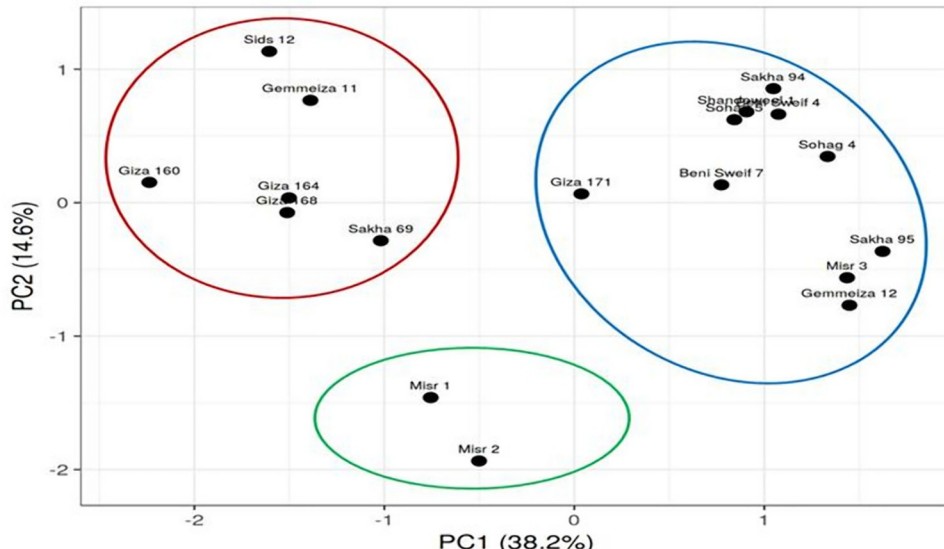

**Fig 2. Principal component analysis (PCA) based on the rust races scoring data.** The figure demonstrates a sharp clustering into three distinctive groups (blue) resistance to stem rust in the seedling stage; (green) moderate susceptible to stem rust races in the seedling stage; genotypes; (red) highly susceptible to stem rust races in seedling stage genotypes; wheat genotypes).

genotypes and ranged from 0 to 80 S for stem rust and from TrMr to 70 S for leaf rust. The highest final stem rust severity percent was recorded for Giza 160, Giza 164, Misr 2, Sakha 69 and Misr1genotypes either at Giza or Sids, but the lowest final stem rust severity% was noticed.

**Table 4. These seedling reaction of 18 Egyptian wheat cultivars tested against 13 races of *P. graminis tritici* under greenhouse conditions.**

| Wheat Cultivar | *P. graminis tritici* | | | | | | | | | | | | |
|---|---|---|---|---|---|---|---|---|---|---|---|---|---|
| | BCQN | CFSC | CFTS | HCCG | KKTS | MMDS | MKKS | MTTN | MTKT | PTTS | PTSS | THTS | TMJT |
| Gemmeiza 11 | L | L | L | | L | | L | L | L | L | L | L | L |
| Gemmeiza 12 | L | L | | L | L | L | | | | | L | L | L |
| Sids 12 | L | L | L | L | L | L | | | L | | L | L | L |
| Misr 1 | | L | | | | | | | | | L | L | |
| Misr 2 | L | L | | L | L | | | | | | L | L | L |
| Misr 3 | L | L | L | | L | | L | L | L | L | L | L | L |
| Giza 164 | | L | | | | | | | | | L | L | |
| Giza 168 | L | L | | L | L | L | | | | | L | L | L |
| Giza 171 | L | L | L | L | L | L | L | L | L | L | L | L | L |
| Sakha 69 | L | L | L | L | L | L | | L | L | | L | L | L |
| Sakha 94 | L | L | L | L | L | L | L | | | | L | L | L |
| Sakha 95 | L | L | L | L | L | L | | L | | L | L | L | L |
| Shandaweel 1 | L | L | | L | L | | | | | | L | L | L |
| Beni Sweif 4 | | L | L | L | L | | | | | L | | | L |
| Beni Sweif 7 | L | L | | L | | | | L | L | L | L | L | L |
| Sohag 4 | | L | | | | L | | | | | L | | |
| Sohag 5 | | L | L | | L | | L | L | | L | | L | L |
| Giza 160 | | L | | | | | | | | | L | | L |

L = low infection type, Blank: High infection type.

**Table 5. Seedling reaction of 18 monogenic lines (*Sr's*) tested against 13 races of *Puccinia graminis f.sp. tritici* under greenhouse conditions in 2019/2020.**

| *Sr's* | BGQN | CBSC | HFTS | LCCG | LTTS | MKKS MCDS | MKKS MCDS | MKTN | MLKT | PFKS | PSSS | PTTS | TTJT |
|--------|------|------|------|------|------|-----------|-----------|------|------|------|------|------|------|
| *Sr5*  | L | L | L |   |   |   |   |   |   |   |   |   |   |
| *Sr10* | L | L |   | L | L | L | L | L | L | L | L | L |   |
| *Sr36* | L |   | L | L |   | L | L | L |   | L | L | L |   |
| *Sr9a* | L | L | L | L | L | L | L | L | L |   |   |   |   |
| *Sr11* | L |   |   | L | L |   |   |   |   |   |   |   |   |
| *Sr6*  |   |   |   | L |   | L | L |   | L | L |   |   | L |
| *Sr8a* | L | L | L | L |   | L | L | L |   | L |   |   |   |
| *Sr9g* |   | L |   | L |   |   |   |   |   |   |   |   |   |
| *Sr26* | L | L |   |   |   |   |   | L |   |   |   |   |   |
| *Sr9b* | L | L | L |   | L | L | L | L |   |   | L | L | L |
| *Sr30* | L | L |   | L |   | L | L |   |   |   |   |   |   |
| *Sr17* | L | L |   |   |   |   |   |   | L |   | L |   |   |
| *Sr28* |   | L |   | L | L | L | L |   |   | L |   | L | L |
| *Sr29* |   |   |   |   |   | L | L |   |   |   |   |   |   |
| *Sr30* | L | L | L |   |   | L | L | L | L | L | L | L | L |
| *Sr32* |   | L |   | L | L | L | L | L | L |   |   | L | L |
| *Sr33* |   | L |   | L | L | L | L |   |   |   | L | L | L |
| *Sr34* | L | L | L |   |   | L | L | L | L | L | L | L |   |

L = low infection type, Blank: High infection type.

While, the area under disease progress curve (AUDPC), as a useful and more reliable quantitative estimator of plant disease severity over time, was also considered to evaluate the 18 genotypes for resistance to stem and leaf rust in Giza and Sids stations. The result revealed that the AUDPC values differed among the tested genotypes. AUDPC values more than 250 were classified as highly susceptible to rusts but those with less than 250 were classified as highly resistant genotypes. Therefore the genotypes Giza 164, Sakha 69 and Giza 160 were highly susceptible to stem and leaf rust in Giza and Sids stations with highly AUDPC values. Misr 1 and Misr 2 genotypes were highly susceptible to stem rust but not to leaf rust. Beni Sweif 4 and Gemmeiza 11 were highly susceptible to leaf rust but not to stem rust. Sids 12 and Sakha 95 were highly susceptible to stem rust in Sids station only. The remaining genotypes displayed the lowest AUDPC values (less than 250) either for stem or leaf rust in both locations (Table 6).

On the other hand, each of the stem rust and leaf rust disease progressed more slowly and increased at relatively lower rates (R-value) on all tested wheat cultivars. According to this the third parameter. The first group included eleven wheat varieties, (Gemmeiza 12, Misr 3, Giza 168, Giza 171, Sakha 94, Sakha 95, Shandaweel 1, Bany Swif 7, Bany Swif 4, Sohag 4 and Sohag 5). These genotypes exhibited a slower rate of stem rust development but increased at relatively lower rates of disease increase (R-values) that ranged from 0.00 to 0.100 during the growing season. The second group included the wheat varieties (Gemmeiza 11, Sids 12, Misr 1, Misr 2, Giza 164, Sakha 69 and Giza 160) which showed the lowest levels of resistance to stem rust infection in comparison with the studied other cultivars as the R-values reached the maximum level, more than 0.100. Therefore, the genotypes were classified as fast rusting or highly susceptible wheat ones. While leaf rust disease severity in the wheat cultivars can also be ranked into

**Table 6. Final stem rust severity (FRS %), area under disease progress curve (AUDPC) and rate of rust disease increase (r-value) of 18 wheat genotypes grown at Giza and Sids Agric. Res. Station, during 2020/21 growing seasons.**

| Genotype | FRS | | | | AUDPC | | | | r-value | | | |
|---|---|---|---|---|---|---|---|---|---|---|---|---|
| | SR | | LR | | SR | | LR | | SR | | LR | |
| | Giza | Sids | Giza | Sids | Giza | Sids | Giza | Sids | Giza | Sids | Giza | Sids |
| Gemmeiza 11 | 20 MS | 10 MS | 40 MS | 50 MS | 143.50 | 87.50 | 283.50 | 297.50 | 0.183 | 0.149 | 0.121 | 0.121 |
| Gemmeiza 12 | Tr MS | 5 MS | 30 MR | 20 MR | 24.50 | 59.50 | 199.50 | 164.50 | 0.067 | 0.067 | 0.069 | 0.040 |
| Sids 12 | 20 S | 40 S | 20 MR | 20 MR | 178.50 | 525.00 | 101.50 | 164.50 | 0.170 | 0.185 | 0.140 | 0.140 |
| Misr 1 | 40 S | 60 S | TrMR | 5 MR | 560.00 | 1050.00 | 45.50 | 66.50 | 0.192 | 0.189 | 0.067 | 0.067 |
| Misr 2 | 60 S | 60 S | 5 MR | 5 MR | 735.00 | 980.00 | 42.00 | 45.50 | 0.215 | 0.283 | 0.053 | 0.067 |
| Misr 3 | Tr S | 5 S | 10 MR | 10 MR | 31.50 | 63.00 | 38.50 | 66.50 | 0.079 | 0.079 | 0.088 | 0.103 |
| Giza 168 | 5 S | 10 S | 20 M | 30 M | 63.00 | 122.50 | 94.50 | 171.50 | 0.104 | 0.036 | 0.032 | 0.007 |
| Giza 171 | 10 M | 0 | 10 R | 20 R | 66.50 | 0.00 | 63.00 | 164.50 | 0.103 | 0.00 | 0.096 | 0.140 |
| Sakha 94 | 5 MS | 20 MS | 5 R | 10 R | 52.50 | 52.50 | 52.50 | 101.50 | 0.067 | 0.067 | 0.067 | 0.088 |
| Sakha 95 | 20 S | 30 S | 10 R | 20 R | 178.50 | 318.50 | 42.00 | 171.50 | 0.100 | 0.108 | 0.096 | 0.032 |
| Shandaweel 1 | 10 MR | 5 MR | 20 MS | 30 MS | 66.50 | 52.50 | 129.50 | 227.50 | 0.100 | 0.067 | 0.132 | 0.164 |
| Giza 164 | 70 S | 60 S | 60 S | 70 S | 910.00 | 1050.00 | 735.00 | 910.00 | 0.245 | 0.285 | 0.224 | 0.245 |
| Sakha 69 | 50 S | 40 S | 70 S | 70 S | 700.00 | 507.50 | 822.50 | 962.50 | 0.205 | 0.221 | 0.281 | 0.281 |
| Beni Sweif 7 | 30 MS | 10 MS | 10 S | 20 S | 234.50 | 115.50 | 80.50 | 178.50 | 0.094 | 0.093 | 0.114 | 0.103 |
| Beni Sweif 4 | 20 MS | 5 MS | 30 S | 40 S | 164.50 | 31.50 | 437.50 | 507.50 | 0.040 | 0.067 | 0.100 | 0.121 |
| Sohag 4 | 10 MS | Tr MS | 20 S | 30 S | 56.00 | 52.50 | 178.50 | 297.50 | 0.079 | 0.067 | 0.153 | 0.100 |
| Sohag 5 | 5 S | Tr S | 20 S | 30 S | 94.50 | 52.50 | 262.50 | 283.50 | 0.032 | 0.067 | 0.074 | 0.178 |
| Giza 160 | 80 S | 80 S | 50 S | 60 S | 1260.00 | 1260.00 | 560.00 | 910.00 | 0.232 | 0.232 | 0.205 | 0.205 |
| Mean | - | - | - | - | 306.639 | 354.472 | 231.58 | 316.167 | 0.128 | 0.126 | 0.117 | 0.122 |
| LSD 0.05 | - | - | - | - | 2.25 | 3.75 | 2.25 | 3.79 | 1.85 | 1.67 | 0.156 | 0.186 |

two main groups. The first group included eleven wheat cultivars, (Gemmeiza 12, Misr 1, Misr 2, Misr 3,Giza 168, Giza 171, Sakha 94, Sakha 95 and Sohag 5) which exhibited stem rust developed more slowly and increased at relatively lower rates of disease increase (R-value). These varieties were designated as the partial resistance and slow rusting ones. Since they displayed the highest level of adult plant resistance or field resistance under the stress of leaf rust infection, they may have durable resistance to leaf rust.

## 3.2. Genetic diversity of the tested wheat genotypes using SCoT and SRAP analysis

**3.2.1. SCoT markers diversity pattern.** In the present investigation, 20 SCoT primers were selected to study the genetic diversity analysis among the eighteen Egyptian wheat genotypes, including stem and leaf resistant and susceptible species. Only eleven SCoT primers generated distinct polymorphic bands. However, a total of 216 bands were polymorphic (97%), and their sizes ranged from 200 to 3000 bp. The number of polymorphic bands for each primer ranged from 6 (SCot71) to 24 (SCot13) and the number of monomorphic bands ranged from 1 (SCoT 14, 24, and 71) to 2 (SCoT 33). Moreover, the number of unique bands ranged from 1 (SCoT 70) to 23 (SCoT 13) as shown in Table 7 and Fig 3. The Jaccard's genetic similarity values of the eighteen Egyptian genotypes, including stem and leaf resistant and susceptible, depend on SCoT molecular markers, ranging from 38 to 84%. The highest genetic similarity revealed by the SCoT molecular markers analysis was 84% recorded between Sakha 69 and Giza 164, which are categorized as susceptible to stem and leaf rust. Meanwhile, the lowest similarity percentage (38%) was recorded for Shandaweel 1 and Gemmeiza 11, given that

**Table 7. Primer code, Size of bands, primer sequence, percentage of polymorphism, total polymorphic and unique bands in the profile of SCoT primer.**

| Primer Code | Size of bands (bp) | Primer sequence | Number of bands | | | Unique band | Polymorphism % |
|---|---|---|---|---|---|---|---|
| | | | Total | Polymorphic | Monomorphic | | |
| SCOT13 | 700–1.900 | 5' ACGACATGGCGACCATCG3' | 47 | 24 | - | 23 | 100 |
| SCOT14 | 500–3000 | 5'ACGACATGGCGACCACGG3' | 20 | 13 | 1 | 6 | 95 |
| SCOT24 | 500–1900 | 5'CACCATGGCTACCACCAT3' | 19 | 12 | 1 | 6 | 94.7 |
| SCOT26 | 300–2500 | 5'ACCATGGCTACCACCGTC3' | 20 | 15 | - | 5 | 100 |
| SCOT31 | 300–1800 | 5'CCATGGCTACCACCGCCT3' | 22 | 15 | - | 7 | 100 |
| SCOT33 | 200–1200 | 5'CCATGGCTACCACCGCAG3' | 20 | 11 | 2 | 7 | 90 |
| SCOT34 | 500–1900 | 5'ACCATGGCTACCACCGCA3' | 26 | 18 | - | 8 | 100 |
| SCOT52 | 200–1100 | 5'ACAATGGCTACCACTGCA3' | 13 | 10 | - | 3 | 100 |
| SCOT61 | 300–2700 | 5'CAACAATGGCTACCACCG3' | 11 | 8 | - | 3 | 100 |
| SCOT70 | 200–1200 | 5'ACCATGGCTACCAGCGCG3' | 9 | 8 | - | 1 | 100 |
| SCOT71 | 300–1700 | 5'CCATGGCTACCACCGCCG3' | 9 | 6 | 1 | 2 | 88.8 |
| Total Average | 200–3000 | | 216 19 | 140 12 | | 71 6 | 97 |

Shandweel 1 is resistant to stem rust but susceptible to leaf rust. While Gemmeiza 11 is categorized as susceptible to both stem and leaf rust (S3 Table). A dendrogram depends on UPGMA analysis using SCoT marker consisting of two main clusters as presented in Fig 5A. The first cluster grouped all plants from the phylogenetically similar susceptible to stem and leaf rust races, i.e. Beni Sweif 7 and Sohag 4. While the second cluster consists of two sub clusters. The second sub cluster I indicated that the most susceptible genotypes to leaf and stem rust (Giza 164, Sakha 69, Sohag 5, Bani Sweif 4, Giza 160 and Shandaweel 1) were genetically similar to three resistant genotypes to leaf rust races (Giza 171, Sakha 94 and Sakha 95). In addition, the second sub cluster II showed that all species belonging to the most resistant genotypes to leaf rust (Misr 1, Misr 2, Misr 3, Gemmiza 12, and Sids12 were similar to some species belonging to susceptible genotypes to stem and leaf rust (Giza 168 and Gemmiza11) races. These results

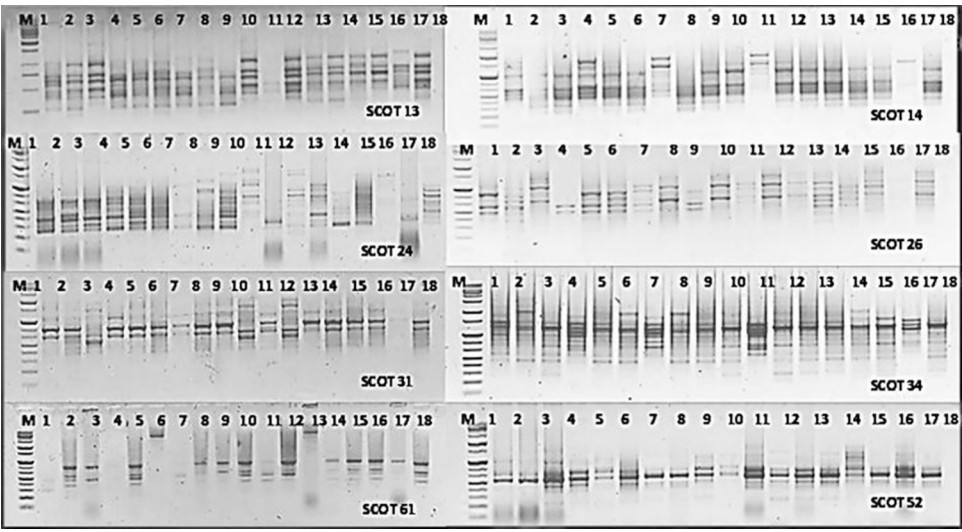

**Fig 3. Amplification profile obtained with eleven ScoT primers in eighteen wheat genotypes.** (M) molecular marker (100 bp); lanes 1–18.

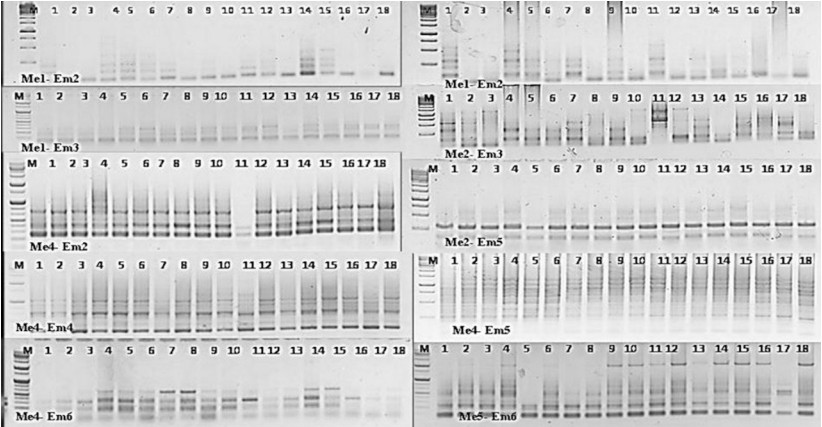

**Fig 4. Amplification profile generated by 14 SRAP primers in 18 wheat genotypes.** (M) molecular marker (100 bp); lanes 1–18.

indicate that SCoT primers have high amplification efficiency and are reliable in the discovery of polymorphisms between stem and leaf rust races.

**3.2.2. SRAP markers diversity pattern.** To investigate the genetic diversity and to evaluate the polymorphism degree among the eighteen Egyptian genotypes, including stem and leaf resistant and also susceptible species. A total of 20 SARP primer combinations were tested against these species genotypes. Only 14 out of 20 primer combinations revealed discernible polymorphism. However, a total of 230 bands were polymorphic (99%), and their sizes ranged from 100 to 2000 bps. The results showed that the total number of bands produced were distinct sharp bands. The number of polymorphic bands per primer combination ranged from 3 (Me1-Em6) to 16 (Me2-Em1), and the number of unique bands per primer ranged from 2 (Me4-Em4) to 19 (Me1-Em3) as presented in Table 8 and Fig 4. The Jaccard's genetic

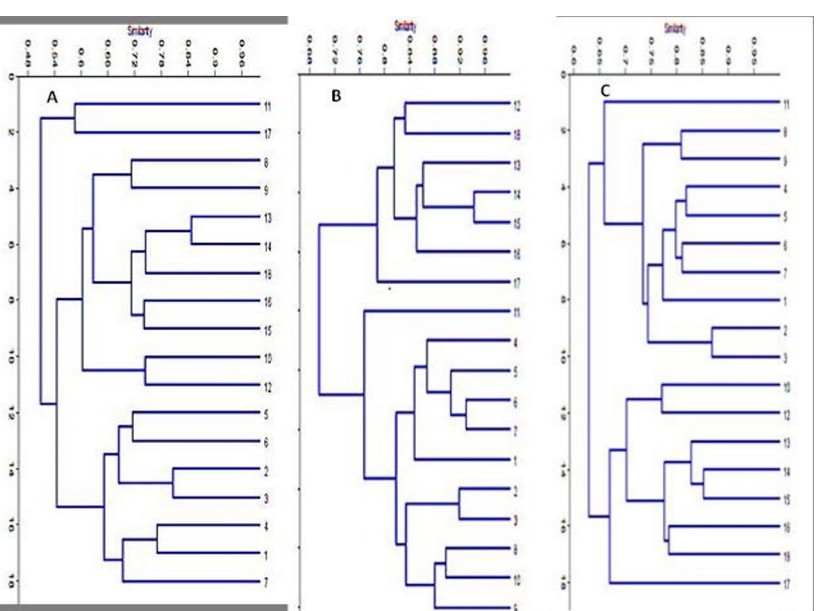

**Fig 5. A dendrogram based on UPGMA analysis using.** (A) SCoT marker, (B) SRAP, (C) Combined data among 18 wheat genotypes.

**Table 8. Primer code, primer sequence, Size of bands and combination total bands, polymorphic bands, unique band and percentage of polymorphism in the SRAP primer.**

| Primer Code SRAP | Size of bands (bp) | Primer combination | Primer sequence | Number of bands Total Polymorphic Monomorphic | | | Unique band | Polymorphism % |
|---|---|---|---|---|---|---|---|---|
| Me1 | 500–1600 | Me1- Em2 | TGAGTCCAAACCGGATA | 20 | 11 | - | 9 | 100 |
| Me2 | 300–600 | Me1- Em3 | TGAGTCCAAACCGGAGC | 28 | 9 | - | 19 | 100 |
| Me3 | 300–600 | Me1- Em4 | TGAGTCCAAACCGGAAT | 16 | 10 | - | 6 | 100 |
| Me4 | 100–800 | Me1- Em5 | TGAGTCCAAACCGGACC- | 17 | 6 | - | 11 | 100 |
| Me5 | 100–2000 | Me1- Em6 | TGAGTCCAAACCGGAAG | 7 | 3 | - | 4 | 100 |
| Em1 | 100–1300 | Me2- Em1 | GACTGCGTACGAATTAAT | 25 | 16 | - | 9 | 100 |
| Em2 | 90–600 | Me2- Em3 | GACTGCGTACGAATTTGC | 8 | 4 | 1 | 3 | 87.5 |
| Em3 | 200–1400 | Me2- Em5 | GACTGCGTACGAATTGAC | 18 | 11 | - | 7 | 100 |
| Em4 | 200–2200 | Me4- Em2 | GACTGCGTACGAATTTGA | 19 | 8 | - | 11 | 100 |
| Em5 | 200–2200 | Me4- Em4 | GACTGCGTACGAATTAAC | 16 | 10 | 4 | 2 | 75 |
| Em6 | 150–1700 | Me4- Em5 | GACTGCGTACGAATTGCA | 7 | 6 | 1 | - | 85.7 |
|  | 100–600 | Me4- Em6 |  | 6 | 6 | - | - | 100 |
|  | 90–200 | Me5- Em2 |  | 11 | 7 | 4 | - | 63.6 |
|  | 170–1500 | Me5- Em5 |  | 12 | 9 | 3 | - | 75 |
|  | 200–1000 | Me5- Em6 |  | 20 | 11 | - | - | 100 |
|  | 100–2000 |  | Total Average | 230 16 | 121 8 | 13 6 | 74 5 | 99 |

similarity values of the eighteen wheat species depend on the SRAP profile molecular markers that ranged from 60 to 94%. The highest genetic similarity value revealed by the SRAP molecular markers analysis was 94% between Sakha 69 and Giza 160, belonging to susceptible genotypes for the stem rust race. Meanwhile, the lowest similarity percentage (0.60%) was noticed for Sohag 5 and Gemmeiza 11 (S4 Table). A dendrogram analysis generated from SRAP analysis based on UPGMA clustering grouped the 18 wheat genotypes into two main clusters as presented Fig 5B. The first cluster consists of the most susceptible genotypes to stem and leaf rust (Shandaweel 1, Sohag 5, Giza 164, Sakha 69, Giza 160, Beni Sweif 4 and Sohag 4). The second cluster grouped the largest number of the most resistant genotypes to leaf rust (Misr 1, Misr 2, Misr 3, Gemmeiza 12, Giza genotypes 171, Sakha 95 and Sakha 94) beside some cultivars susceptible to stem and leaf rust (Beni Sweif 7, Giza 168, Gemmeiza 11, Sids 12). According to the cluster analysis, the SRAP data successfully clustered similar resistant and susceptible genotypes to leaf and stem rust into the same phenotypes and genotypic groups. These results show that the SRAP clusters exhibited a relatively direct connection with plant sequence taxonomy and assessment of stem and leaf rust races.

**3.2.3 Diversity analysis using the combined data.** The generic dendrogram was constructed using the combined data of all the molecular markers used in this investigation (SCoT and SRAP) grouped the eighteen studied wheat genotypes into two main clusters as presented in Fig 5C. Cluster 1 comprises of the most resistant genotypes to the leaf rust pathogen i.e. Beni Sweif 7, Giza 171, Sakha 94, Misr 1, Misr 2, Misr 3, Giza 168, Gemmeiza 11, Gemmeiza 12 and Sids 12. Meanwhile, Cluster II consists of the most susceptible genotypes to both stem and leaf rust (Sakha 95, Shandaweel 1, Giza 164, Sakha 69, Giza 160, Beni Sweif 4, Sohag 5 and Sohag 4). The data scored from SCoT and SRAPP were combined and analyzed to produce the deeper relationships dependent on the wider and more versatile genome coverage. The combined dendrogram included two clusters with a high topology that harmonized with the SRAP dendrogram only. Finally, the Jaccard's genetic similarity values of the eighteen wheat species

depend on the combined data and displayed consistent results that were comparable to the grouping produced by the cluster analysis. The combined data analysis results confirmed the presence of the highest genetic similarity (85%) between Sakha 69 and Giza 160 as presented in S5 Table.

## 3.3. Characterization of nanoparticles using Transmission Electron Microscopy (TEM)

**3.3.1. Effect of Cu-chitosan composite nanoparticle treatments and the application methods on incubation period, latent period and infection type of wheat stem rust pathogen.** In our earlier study, these CuChNp were well characterized for various physico-chemical properties, like the interaction of chitosan with Cu and the internal architecture by transmission electron microscopy (TEM) (Fig 6). In the present investigation, the eighteen wheat genotypes were treated with cu-chitosan composite nanoparticles either before or after inoculation with uridiospores of the stem rust pathogen to study and determine the effect of nanoparticle treatment and its application methods in controlling the disease. The estimation was expressed as incubation, latent period stem and leaf rust infection type as presented in Tables 9 and 10. The results of CuChNp treatment on stem and leaf rust races revealed that the infection process was reduced when the plant was treated 24 h before and 24 h before and after the inoculation. The incubation and latent periods were increased in treated wheat plants of the tested genotypes rather than in the untreated plants. Besides, the treatment gave the lowest infection type compared to the control ones.

## 3.4 *In silico* interpretation of the Cu-chitosan composite nanoparticle against stem rust and leaf rust pathogens

**3.4.1 3D structure prediction, quality assessment and validation.** BLASTp analysis provided PDB ids: 2VKN and 2FA2 putative templates, respectively showed high-level sequence identity with MAPK1 and PgMAPK sequences as presented in Table 10. Accordingly, the BLAST analysis PDB ID: 2vkn chain A with a resolution of 2.05 Å and 2FA2 chain A with a

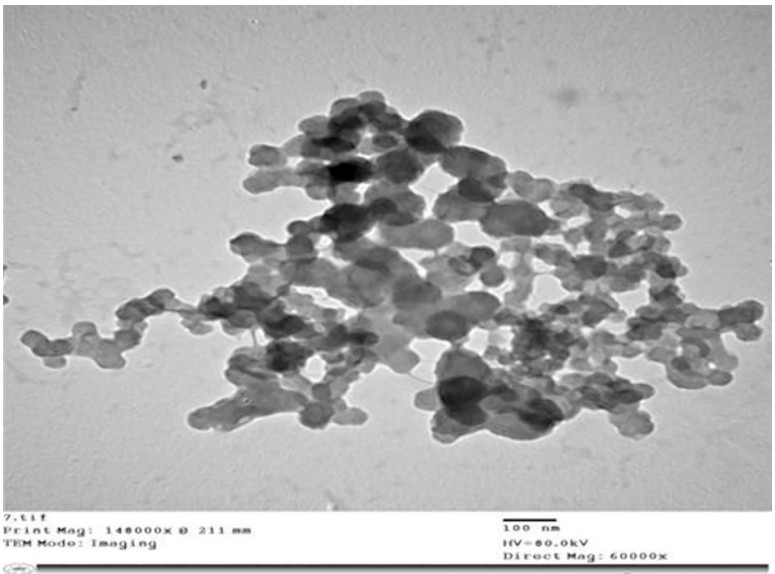

**Fig 6. EM micrographs of the aggregated Cu–chitosan nanoparticles composite.**

**Table 9. Effect of Cu-chitosan NPs treatments and application methods on incubation, latent periods and infection type of the wheat stem rust disease.**

| Treatment/ Verities | Incubation period/day | | | Latent period/day | | | Infection type | | |
|---|---|---|---|---|---|---|---|---|---|
| | Control | Before* | Post** | Control | Before* | Post** | Control | Pre | post |
| Gemmeiza 11 | 8.59 | 14.3 | 15.04 | 12.04 | 19.95 | 19.97 | 3 | 2+ | 2+ |
| Gemmeiza 12 | 7.23 | 16.27 | 15.54 | 13.28 | 20.05 | 20.31 | 2 | 1+ | 1 |
| Sids 12 | 6.98 | 14.21 | 14.56 | 13.46 | 19.92 | 19.35 | 4 | 2++ | 2+ |
| Misr 1 | 5.67 | 13.16 | 13.45 | 11.67 | 18.67 | 18.67 | 4 | 2+ | 2+ |
| Misr 2 | 5.76 | 13.33 | 13.57 | 11.06 | 20.21 | 20.31 | 3 | 2++ | 2+ |
| Misr 3 | 7.85 | 16.83 | 16.92 | 13.45 | 21.22 | 21.04 | 2 | 1 | 1 |
| Giza 164 | 6.03 | 13.57 | 13.89 | 13.78 | 19.92 | 20.81 | 4 | 2++ | 3 |
| Giza 168 | 7.07 | 15.76 | 15.78 | 12.86 | 20.05 | 20.09 | 3 | 1 | 2 |
| Giza 171 | 8.45 | 15.37 | 15.62 | 12.35 | 19.07 | 19.08 | 2 | 1 | 1 |
| Sakha 69 | 7.63 | 14.56 | 14.87 | 12.35 | 19.09 | 20.18 | 3 | 2 | 1++ |
| Sakha 94 | 8.12 | 16.93 | 16.36 | 11.92 | 19.92 | 19.94 | 2 | 1 | 1 |
| Sakha 95 | 7.95 | 17.56 | 17.45 | 13.00 | 20.63 | 20.97 | 3 | 1 | 1+ |
| Shandaweel 1 | 8.31 | 17.93 | 17.97 | 13.57 | 20.46 | 20.93 | 3 | 2 | 2+ |
| Beni Sweif 4 | 8.34 | 17.84 | 17.92 | 14.01 | 20.23 | 22.03 | 2 | 1 | 2 |
| Beni Sweif 7 | 7.93 | 17.36 | 17.35 | 12.04 | 19.07 | 19.04 | 2 | 1+ | 1 |
| Sohag 4 | 6.38 | 15.93 | 15.38 | 13.75 | 19.2 | 19.83 | 2 | 1+ | 1 |
| Sohag 5 | 7.01 | 16.26 | 16.17 | 13.84 | 19.35 | 19.47 | 2 | 1 | 1 |
| Giza 160 | 5.01 | 12.35 | 12.48 | 11.87 | 19.08 | 19.09 | 4 | 3 | 3+ |

* spray before inoculation by 24 hours

**Spray before and after inoculation by 24 hours.

0, 1,1+, 2, 2+, 2++ (Low infection types) Resistance.

3. 4 (High infection types) Susceptible.

resolution of 2.85 Å reflected the best template structure for the comparative model building of MAPK1 and PgMAPK, respectively. The query coverage of protein sequences revealed MAP kinase 1 of *P. triticina* and PGTG of *P. graminis f. sp. tritici* (84 and 12%) query coverage's with 56.14, 56.15 identity with the template proteins (2VKN, 2FA2) that used as template proteins for homology modeling of our target proteins. The SWISSMODLE server-generated 25 and 35 predictive models for MAPK1 of *P. triticina* and PgMAPK of *P. graminis* f. sp. *tritici* proteins with different (QMEAN) score values. The model with low values for QMEAN score (-1.53 and -0.69, respectively) was selected as a final model for *in silico* characterization and docking studies.

The structural analysis and verification server were used to analyze and validate the stability of the MAPK1 and PgMAPK models (https://services.mbi.ucla.edu/SAVES/). The reliability of the backbone of torsion angles i.e. φ and Ψ was evaluated using the PROCHECK program, which measures amino acid residues falling in the existing regions of Ramachandran plot, as depicted in Fig 7A and 7B. Analysis of Ramachandran plot for MAPK1 and PgMAPK revealed that, respectively 91 and 93% residues were found in the most favored regions (A, B, and L) and only 8.7 and 6.1% residues occupied the additionally, allowed regions (a, b, l, and p). 8.3% residues in additional acceptable regions, 0.8% residues in generously acceptable regions, and 1.4% residue in disallowed regions (S6 Table). The quality of the MAPK1 model was further justified by a better ERRAT score of 71.6931 (a value of ~ 95% reveals high resolution), which recommended an acceptable environment for protein (Colovos and Yeates, 1993). This confirms that the predicted model quality MAPK1 of *P. triticina* and PgMAPK of *P. graminis* f. sp.

**Table 10. Effect of Cu-chitosan NPs treatments and application methods on incubation, latent periods and infection type of the wheat leaf rust disease.**

| Treatment/ Verities | Incubation period/day | | | Latent period/day | | | Infection type | | |
|---|---|---|---|---|---|---|---|---|---|
| | Control | Before* | Post** | Control | Before* | Post** | Control | Pre | post |
| Gemmeiza 11 | 7.21 | 13.45 | 13.99 | 10.25 | 18.95 | 18.81 | 2++ | 0 | 1 |
| Gemmeiza 12 | 7.21 | 15.23 | 16.34 | 12.22 | 19.22 | 16.21 | 2+ | 1 | 1 |
| Sids 12 | 7.23 | 15.22 | 16.55 | 12.45 | 18.65 | 18.33 | 3 | 2+ | 2 |
| Misr 1 | 6.76 | 12.22 | 12.73 | 10.64 | 17.45 | 18.43 | 2 | 1 | 1 |
| Misr 2 | 6.63 | 12.23 | 12.42 | 10.11 | 18.22 | 18.23 | 2 | 1 | 1 |
| Misr 3 | 5.55 | 15.42 | 16.53 | 12.12 | 19.20 | 19.04 | 2 | 0 | 1 |
| Giza 164 | 5.22 | 12.52 | 12.90 | 10.76 | 16.43 | 19.18 | 4 | 2 | 2 |
| Giza 168 | 6.11 | 12.34 | 16.47 | 11.87 | 17.16 | 19.21 | 4 | 2 | 2 |
| Giza 171 | 6.98 | 16.32 | 16.63 | 10.33 | 17.09 | 18.13 | 2 | 1 | 1 |
| Sakha 69 | 6.31 | 13.55 | 16.77 | 11.35 | 16.17 | 17.22 | 4 | 2 | 2 |
| Sakha 94 | 6.23 | 17.90 | 18.33 | 10.92 | 18.92 | 17.86 | 3+ | 2 | 2+ |
| Sakha 95 | 8.75 | 16.55 | 18.55 | 10.11 | 18.22 | 21.64 | 2 | 0 | 1 |
| Shandaweel 1 | 7.32 | 15.23 | 18.23 | 13.53 | 21.44 | 22.53 | 3 | 1 | 2 |
| Beni Sweif 4 | 7.433 | 18.85 | 18.93 | 11.03 | 16.24 | 19.14 | 3 | 1 | 2 |
| Beni Sweif 7 | 6.22 | 14.33 | 15.45 | 10.13 | 18.21 | 18.13 | 4 | 1 | 2 |
| Sohag 4 | 7.31 | 14.81 | 14.99 | 12.26 | 17.22 | 18.74 | 3 | 1 | 1+ |
| Sohag 5 | 6.21 | 15.22 | 16.18 | 11.44 | 17.36 | 18.42 | 3 | 1 | 2 |
| Giza 160 | 7.12 | 13.54 | 13.56 | 10.21 | 18.12 | 18.95 | 4 | 2 | 2+ |

* spray before inoculation by 24 hours

**Spray before and after inoculation by 24 hours.

0, 1,1+, 2, 2+, 2++ (Low infection types) Resistance.

3. 4 (High infection types) Susceptible.

*tritici* had a good stereochemical quality and was close to the template structure. The ProQ is a neural network-based predictor based on several of structural features as it predicts the quality of the protein model. The ERRAT score for the modeled structure was found to be 96.22 and 78.4314, respectively. The 3D revealed that the predicted proteins MAPK1 and PgMAPK have 96.22 and 81.36% of the residues have an average 3D-1D score > = 0.2 Pass At least 80% of the amino acids have scored > = 0.2 in the 3D/1D profile. Moreover, the ligand-binding sites identified in the target modeling proteins structure are presented in Table 11 and Fig 8A and 8B.

**3.4.2 Protein-ligand docking.** In this investigation, the modeled proteins MAPK1 and PgMAPK were docked with all the chitosan and chitosan- copper complex to generate their binding mode to refine the best the pose with an allowed conformational change in the tested proteins. The MPAK1 was bound to chitosan with ΔGbind were bound and docking score was -6.9 kcal/mol with the interactive binding site residues (ARG106, ASP192, GLU210, GLY212, LEU213, GLU223, THR224, GLY225, MET227, VAL231, ALA232 and ARG234).While, the MPAK1 was bound to chitosan with CU the ΔGbind are -6.6kcal/ mol with the interactive binding site residues (ARG106, LYS194, GLU223, MET227, GLU229, VAL231, ALA232, GLN247 and GLY273). On the other hand, the PGAT protein for leaf rust was bound to chitosan with ΔGbind -4.6kcal/mol with the interactive binding site residues (TYR441, PRO448, ASN449, SER483, SER478, GLN470 and LYS467). Also, the PGAT was bound to chitosan with CU the ΔGbind are -5.6 kcal/mol with the interactive binding site residues (pro 448, Ala443, Glu450, Arg472, Arg 473 and Ph453) as presented in Fig 9A and 9B.

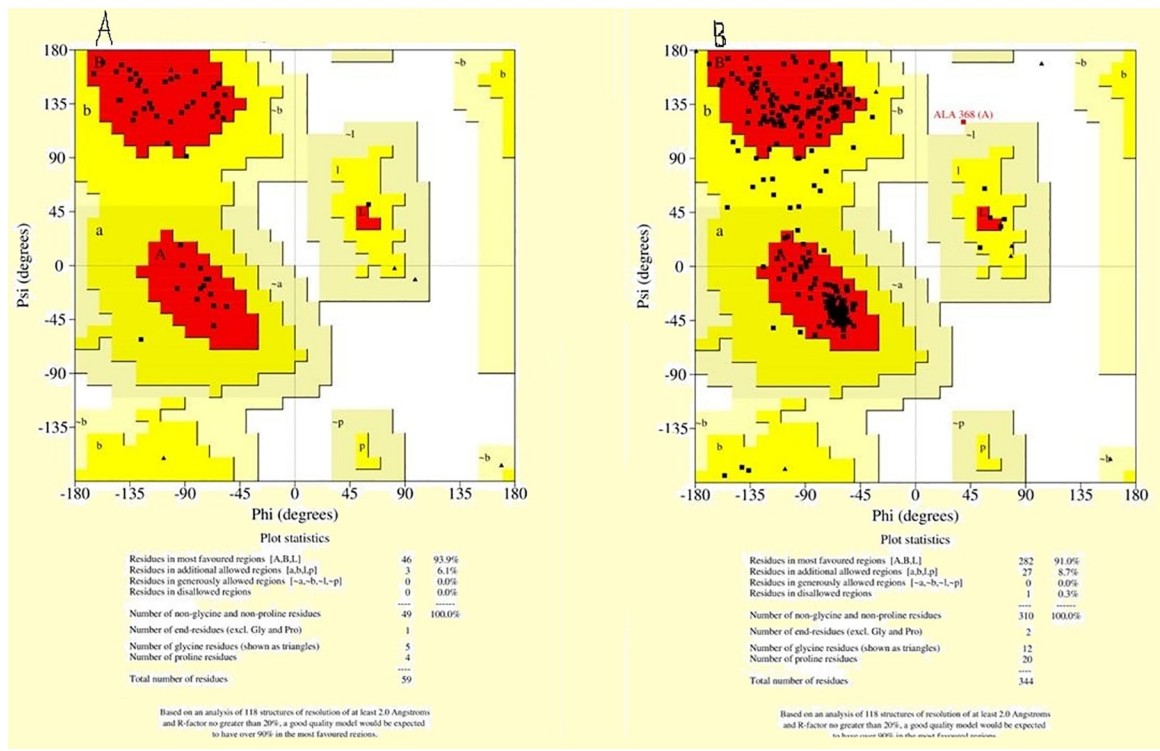

**Fig 7. Ramachandran plot of the modeled structure.** (A) MAPK1 and (B) PgMAPK evaluated and checked by SAVES. For both residues, the plot demonstrates phi-psi torsion angles.

## 4. Discussion

Wheat is one of the major important crops for the urgent need for food supply all over the world. Wheat stem rust (*Puccinia graminis f. sp. tritici*) and leaf rust (*Puccinia triticina*) are the most common rust diseases that constitute the greatest threats that destructively influence plant growth, grain quality, and yield production. There are many approaches to controlling crop diseases, including genetic breeding, cultural sanitation systems, and new pesticide products. Host genetic resistance is the most promising method for the control of plant diseases. Therefore, the present study is designed to analyze the morphological and genetic diversity between susceptible and resistant eighteen Egyptian genotypes to the stem and leaf rust diseases using modern analytical approaches that use functional markers such as SCoT and SRAP. Besides, we characterized and assessed the biological activities of the Cu-chitosan

**Table 11. Binding free energy of chitosan alone or complexed with copper predicted by molecular docking with MAPK1 and PgMAPK proteins of *P. triticina* and *P. graminis tritici*, respectively.**

| Fungi | Grid box (x, y, z) | Protein/accession no. | Ligands | Binding energy |
|---|---|---|---|---|
| *P. triticina* | x = 8.6875<br>y = 7.9130<br>z = 26.0847 | MAP kinase 1 *P. triticina* (AAY89655.1) | chitosan | -6.9 |
| | | | cu-chitosan | -6.6 |
| *P. graminis* f. sp. *tritici* | x = 11.3055<br>y = 22.877<br>z = 1.3282 | PGTG *P. graminis* f. sp. *tritici* (10532062) | chitosan | -4.6 |
| | | | cu-chitosan | -5.6 |

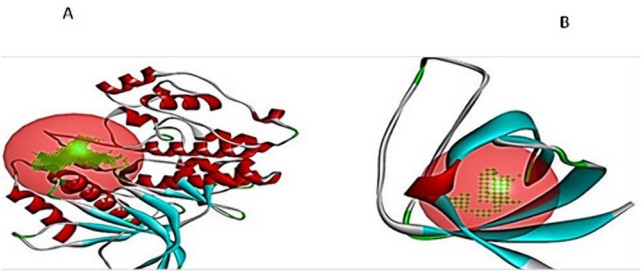

**Fig 8. The binding site of the rust fungus.** The MAKP1 of the leaf rust fungus and (B) the PGTG protein of stem rust.

nanoparticle against stem and leaf rust diseases. Moreover, the molecular docking analysis was also used to study the mode of action of Cu-chitosan nanoparticles in disease control management.

In this regard, Egyptian wheat varieties collected from different regions have a range of resistant genes. They can be used as a new source of rust resistance genes to obtain and grow novel resistance genotypes of wheat [37]. The present study is designed to evaluate the eighteen Egyptian wheat genotypes in the seedling stage against twenty different stem rust races. The experiment was carried out under greenhouse conditions as well as in the adult stage in the field located in two different locations during the 2020 growing season. The results of stem rust disease under greenhouse conditions in the 2020 season revealed that the resistance response was observed in some wheat genotypes while the susceptible reaction was detected in the second group of wheat genotypes and some of the genotypes were susceptible to almost all races. In this study, we used a transcriptomic-derived heat map to globally corroborate Metascape analyses and component analysis (PCA). It successfully separated the wheat genotypes under study into three groups (resistance, susceptible, and highly susceptible). This result was confirmed by the results obtained at the adult plant stage. This result is consistent with [29,38]. On the other hand, the results of the leaf rust disease revealed that the susceptible reaction was

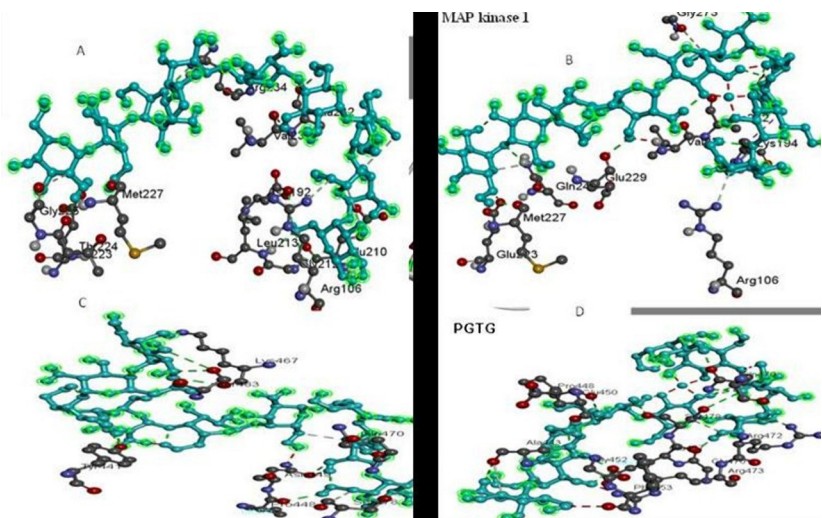

**Fig 9. The overall 3D surface view of the modeled protein.** PtMAPK1 and PGTG with chitosan only and chitosan–Cu complex. MAPK1 with chitosan only, (B) MAPK1 with chitosan–Cu complex, (C) PGTG with chitosan only and, (D (PGTG with chitosan–Cu complex.

identified in 11 genotypes. Whereas, the resistant response was determined in 7 genotypes. Therefore, the apparent disparity between the tested genotypes in their seedling reaction susceptibility to stem rust breeds could have a complex genetic history. Among the 18 genotypes, the high frequency of MS to S genotypes suggests that ineffective stem rust resistance genes are present in their genetic backgrounds. Likewise, within Brazilian genotypes with aluminum tolerance backgrounds, [39] documented a high frequency of moderate response to susceptible infection forms. Besides, 9 and 2 genotypes showed clear and moderate resistance (RMR) to leaf rust and stem rust, respectively. This indicates that these lines could bear various resistance genes that are effective against leaf rust and stem rust and hence could be used effectively as parents in breeding programs. This result was confirmed by those obtained by [40–45].

The gene postulation applies the principles of gene-for-gene theory to theorize which stem and leaf rust resistance genes could possibly be present in the wheat genotype. The effective advantage of the gene postulation test is that it predicts the probability of the presence of resistance genes in a few weeks using the primary leaves of seedling plants. The postulation gene at seedling stage may lead to the conclusion that if any wheat cultivar is proved to have only one single gene for stem or leaf rust resistance, it won't lead to durable resistance due to the rapid development of new physiologic races of the stem and leaf rust fungus. Therefore, this cultivar will be discarded soon after it's released. In this study, cultivars Giza 160 and Giza 164 do not contain any resistance genes of the *Sr, s* set and cultures that were used in this study. These results are in accordance with those previously reported by [40]. On the other hand, some wheat genotypes, like Giza 171, Gemmiza 11 and Gemmiza 12, display many resistance genes. This result was inconsistent with [46]. The reason may be supplanting the breaking of genotype resistance could be attributed to the appearance of new races of the pathogen.

On the other hand, AUDPC values for stem and leaf rust significantly differed among the studied wheat genotypes. However, performance revealed grouping of the 18 wheat varieties into two distinct groups. Wheat genotypes in the first group with values of less than 250 are classified as highly resistant genotypes. The second group displayed high estimates of AUDPC, more than 250 classified as highly susceptible to rust. In this group, the wheat genotypes were divided into two groups, moderately susceptible and highly susceptible. This result was consistent with the results of the heat map and principal component (PCA) as analyses revealed that one-dimensional heatmap visualization of the interaction between the presence of rust races and wheat variety performance revealed grouping of the 18 wheat varieties. These results are in good agreement with those previously reported by [5,29,47–49]

The rate of rust disease increase (r-value) is one of the epidemiological parameters used for quantitative determination of partial resistance to stem and leaf rusts under field conditions. In this study, the performance revealed grouping of the 18 wheat genotypes into two distinct groups: the resistance group, as they displayed lower rates of disease increase with low r-value, and the susceptible group, the fast-rusting or the highly susceptible ones, when subjected to the same pathogen populations and under the same environmental field conditions as in the current study. However, it was previously reported that differences in (r-value) estimates between two wheat cultivars tested during an epidemic development are due to one of two reasons. The first reason is the disease severity in each, and the second reason is the spread of the rust races reproduction, or the time of disease increase [22].

In this investigation, the genetic relationship between different plant species and genera is very important because new relationships between plants need to be discovered in plant evolution [50]. Genetic diversity is a requirement for detecting leaf and stem rust-resistant wheat cultivars [51]. Accordingly, we carried out gene-targeted molecular markers (SCoT) and sequence-related amplified polymorphisms (SRAP) analyses to determine their elevated power in polymorphism identification in wheat. In this study, the SCoT and SRAP markers were

approved to study the genetic diversity among eighteen wheat genotypes, including resistant and susceptible species to stem and leaf rust. The results revealed that the gene-targeted molecular marker (SCoT) has several advantages over the use of dominant random markers (such as ISSR, RAPD, and AFLP). However, these markers reveal genetic diversity from the genic region in the genome and this functional diversity can be used in any species [52]. Moreover, the SRAP marker system is a new, simple, and efficient marker system that can be adapted for a variety of purposes, such as linkage map construction [53], genomic and cDNA fingerprinting gene tagging [24], and genetic diversity analysis. The results revealed that SRAP markers have the potential to improve the current suite of molecular tools in a diversity of fields by providing an easy-to-use, highly variable marker with essential biological significance. Abou-Deif, *et al.*, [54] used the ISSR molecular marker to estimate the genetic diversity between some Egyptian wheat genotypes. But El-Moneim, [55] and Gowayed and El-Moneim [9] used ISSR and SCoT primers to determine and evaluate some Egyptian wheat genotypes tolerant to drought and salt stress, respectively. But our current study used different primers for SCoT in genetic diversity and evaluation of Egyptian wheat genotypes' resistance to stem and leaf rust diseases. The SCoT and SRAP markers indicated the highest level of polymorphism (97 and 99% respectively) and were established to be effective in detecting genetic diversity among the eighteen genotypes studied. Besides, there were a number of 71 and 74 unique loci for SCoT and SRAP molecular markers, respectively. The highest similarity values among the 18 genotypes obtained by SCoT and SRAP primers were noticed between Sakha 69 and Giza 164, which are characterized as susceptible to stem and leaf rust. This could be interpreted that the two genotypes in their history may bear ineffective genes for stem rust and leaf rust resistance. Shadwell 1 (moderately resistant to stem rust) has a genetic background far from Gemmeiza 11 (susceptible to both stem and leaf rust) according to the similarity value (38%) obtained by SCoT analysis, but Shandweel 1 varied with Sohag 5 (susceptible to stem and leaf rust) with a similarity of 0.95 percent according to SRAP analysis. The determined high level of polymorphism was indicative of the greater genetic diversity among the studied genotypes, which can be effectively utilized for gene tagging and genome mapping of crosses to introgress beneficial traits into thecultivated genotypes. Our results agreement with [56] evaluated the genetic relationships of sixteen species belonging to two major genera of Mammillaria and Notocactus in the family Cactaceae using modern gene-targeting marker techniques, i.e. the Start Codon Targeted (SCoT) Polymorphism. Moreover, Hanaa *et.al.*, [57] reported that the gene-targeted marker techniques (SCoT) were highly beneficial tools for the classification of sixteen medicinal plant species. In general, in distinguishing between resistant and susceptible wheat genotypes, especially leaf rust, SRAP was more effective than SCoT, as it showed the greatest genetic polymorphism. In addition, SRAP (Me1-Em2 and Me2-Em5) displayed the largest number of polymorphic bands, but the largest specific bands were given by Me4-Em2 and Me1-Em5 SRAP primers. Thus, the wheat genotypes were considered to be the best primers to determine and distinguish the wheat genotypes. In addition, in both SRAP and combined data analysis, the similarity matrix showed a close relationship between Sakha 69 and Giza 160 genotypes. At the same time, the SRAP clusters had a relatively direct connection with plant wheat taxonomy. The ability of the SRAP marker to distinguish and decide genetic diversity among genotypes is considered to be the most important feature of the best marker [55]. In this respect, [58] reported that several markers, such as inter-simple sequence repeat (ISSR), random amplified polymorphic DNA (RAPD) and amplified fragment length polymorphism (AFLP), have been applied to address hypotheses at lower taxonomic levels in the past few decades. But, lately, sequence-related amplified polymorphism (SRAP) markers have been established, which are done to amplify coding regions of DNA with primers targeting open reading frames. These markers have been confirmed to be robust and highly variable, on

equivalence with AFLP, and are attained through a significantly less technically demanding process. Moreover, the SRAP markers have been used for assessing the genetic diversity of large germplasm collections. In this respect, [59] reported that the Simple Sequences Repeats (SSR) and SRAP were fast, accurate, and high throughput fingerprinting could be acquired using those markers, from the combined analysis, which revealed the existence of significant variation among the 33 accessions. The most distant accessions can be used by breeders to develop improved sorghum genotypes. Therefore, SRAP and SCoT markers may be appropriate for distinguishing the resistance and vulnerability of wheat genotypes to stem and leaf rust as a result of the wide genetic variation observed among the 18 wheat genotypes. On the other hand, the association of molecular markers with stem and leaf rust evaluation is an important factor in understanding the genetic role of tolerance by predicting the genomic regions that affect the plant's response to stem and leaf rust disease.

In this investigation, for the first time, the activity of chitosan nanoparticles (CuChNp) was studied and shown to have the potential to inhibit stem and leaf rust in studied Egyptian wheat genotypes. In this respect, Kheiri *et al.* [60] reported that CS/NPs have important inhibitory effects on the growth of fungi, colony formation and conidial germination of *F. grinearum*. In addition, *in vitro* plates treated with chitosan-Cu nanoparticles showed effectively inhibited mycelial growth and spore germination of *Alternaria alternata* (90%), *Macrophomina phaseolina* (63%), and *Rhizoctonia solanii* (60%) [61]. The antimicrobial effect of chitosan on pathogens (bacteria, fungi and viruses) is based on several mechanisms [62]. The positive charge of protonated chitosan allows electrostatic interactions with the negative charge of the surface of the pathogen [63]. The damage to the cells and the leakage of the pathogen increase the permeability of the membrane, resulting in cell death [32]. Chitosan then chelates the necessary elements (including metal ions, minerals, and nutrients) for the creation of pathogens, thus preventing the normal growth of pathogens [53]. The interaction of pathogens with penetrated chitosan by DNA/RNA contributes to the inhibition of mRNA syncretism and pathogen reproduction and the deposition of chitosan on the microbial surface of pathogens [64]. Finally, chitosan deposition on the microbial surface of pathogens constitutes a barrier to the extracellular transport of critical nutrients and metabolites from entering the cell, thereby inhibiting the normal growth of pathogens [13]. Due to the large surface area that comes in contact with the pathogens, the efficiency of the system in action can be improved by the small size of the chitosan nanoformulations. The small size can also increase the absorption and increase the thick coat of seeds, plant tissues, as well as the cell membranes of pathogens of the penetrated and permeated chitosan, resulting in better plant immunity and defensive response activities. This is the first study to use Cu-chitosan composite nanoparticles in foliar fertilizer applications that could help in maintaining and controlling rust.

Moreover, the activity and mode of action of chitosan-copper composite nanoparticles in controlling the wheat disease stem rust were studied using molecular docking analysis. Computational modeling could determine exceptional information to understand the mechanism of the mode of action of the antifungal molecules that inhibit the fungal infection process. The molecular docking approach was used to predict the molecules that have the ability to bind specifically to the protein active site that is responsible for the fungal infection process 34. In the present study, the docking of the active molecules of the studied chitosan-copper nanoparticles with two essential proteins involved in the stem and leaf rust development pathway was evaluated. In this respect, the results of molecular docking analysis showed that the MPAK1 protein for stem rust was bound to chitosan with a binding affinity of-6.9 kcal/mol, while the MPAK1 was bound to cu-chitosan nanoparticles with a binding affinity of-6.6kcal/mol. On the other hand, the PGAT protein for leaf rust was bound to chitosan with a binding affinity of-4.6kcal/mol, while this protein also bound to cu-chitosan nanoparticles with a

binding affinity of -5.6 kcal/mol. Hence, this leads to inhibition of stem and leaf rust pathogenicity. These proteins are a vital element of the pathway of the transduction signal that controls numerous stem and leaf rust infection processes in all organisms. They are responsible for the phosphorylation of target transcription factors that trigger particular genes [65,66]. In many fungi, molecular genetic studies have revealed that MAPK homologs fall into three major subgroups defined in Saccharomyces cerevisiae based on the following diverse functions: osmoregulation and other stress responses (ScHog1 and YSAPK-like), various developmental processes such as mating and filamentation, hyphal formation, conidiation, and conidial germination (ScFus3/ In pathogenic fungi, mutants tend to affect pathogenesis in homologs of all three classes, which is not surprising given the broad functions they impinge on. The MAPK gene was isolated for the first time from *P. triticina* by Guanggan *et al.*, [67]. This wheat leaf rust gene was called PtMAPK1 and revealed that it codes for a protein that is closely homologous to the Ustilago maydis YERK1 subfamily kinases (Ubc3/Kpp2 and Kpp6). These MAPKs can play a role in mating interactions and invasion of plant tissue and pathogenic growth after mating. Also, it plays an important role in haustorium and infection peg formation [68,69]. Presently, no crystalline structural data has been reported for MPAK1 and PGAT. This information gap hinders the development and improvement of MPAK1 and PGAT modulators. In the present study, 3D MPAK1 and PGAT protein structures were plotted using SWISS-MODEL according to the two 2VKN and 2FA2 putative templates, which have proved to be a powerful and valuable tool for homology modeling [70]. Thus, docking analysis confirmed the ability of the cu-chitosan nanoparticles to inhibit the two rust studied pathogens through their high binding affinity for modelling MAPK machinery, resulting in disturbance of key pathways involved in growth, mating and virulence of the pathogen. This will help in the prevention and management of leaf and stem rust in an eco-friendly manner.

Our findings during this investigation concluded that primers for SRAP (Me1-Em2 and Me2-Em5) and SCoT13 proved to be sufficient for assessing the genetic diversity of the tested wheat genotypes. Thus, to classify the resistance genes, their sequences could be used to identify the parents of wheat genotypes. Besides, spraying foliar wheat plants with the cu-chitosan nanoparticles composite plant reduced the form of host reaction to stem and leaf rusts.

## 5. Conclusion

Wheat production can be enhanced by using tolerant and resistant genotypes. Therefore, evaluation and assessment of the stem and leaf rust resistance potential in wheat through marker analysis could potentially diminish the cost of breeding programs and be a powerful strategy for the selection of the most resistant stem and leaf rust genotype. SARAP markers were more efficient and had stronger discriminating power than SCoT markers for assessing the genetic diversity of the tested wheat genotypes, as shown by the number of specific bands, polymorphism percentage, and high values of the genetic diversity indices. Besides, spraying foliar wheat plants with the cu-chitosan nanoparticles composite plant reduced the form of host reaction to stem and leaf rust and increased incubation and latent periods. An insight into the interactions between cu-chitosan nanoparticles and MAPK *P. graminis tritici* or MAPK *P. triticina* protein is given in the computational analysis. Besides confirming the validity of virtual screening techniques, our predictive and experimental results could provide a valuable starting point for the design of MAPK *P. triticina I* and PGTG of *P. graminis f. sp. tritici* nhibitors are able to limit the transduction signal that controls numerous stem and leaf rust infection processes. It elucidates the way the fungal infection is inhibited by MAPK protein targeting. This is the first report on the use of (CuChNp) of foliar fertilizers for controlling rust disease in wheat. The results showed that (CuChNp) could be a useful biological pesticide for controlling

stem and leaf rust diseases. We proposed that the CuChNp represents a safe and good opportunity for the development of commercial plant protection against rust disease. However, with regard to the antifungal activity of CuChNp, many experimental trials are needed to understand the particular mechanism of penetration and transportation of these treatments into plant cells and their interaction with fungal cells in these cells.

## Supporting information

**S1 Table. A virulence/virulence pattern of 20 *Puccinia graminis f.sp. tritici* pathotypes used to study stem rust resistance of eighteen Egyptian wheat genotypes.**
(DOCX)

**S2 Table. A virulence/virulence, based on seedling reaction for 5 isolates leaf rust in 2019/ 2020 season.**
(DOCX)

**S3 Table. Genetic similarity matrix among the 18 wheat genotypes based on the Dice coefficient generated from SCOT marker.**
(DOCX)

**S4 Table. Genetic similarity matrix among the 18 wheat genotypes based on the Dice coefficient generated from SRAP marker.**
(DOCX)

**S5 Table. Genetic similarity matrix among the 18 wheat genotypes based on the Dice coefficient generated from combined data.**
(DOCX)

**S6 Table. Sequence identity, Query coverage and QMEAN scores of templates structure for homology model building of MAPK1 and PgMAPK proteins of *P. triticina* and *P. graminis tritici* through BLAST against RCSB PDB.**
(DOCX)

## Acknowledgments

We would to thank Israa M. Shamikh the graduated student and technical assistant for her contribution during the time of executing the docking analysis part under the supervision of Hanaa S.omar.

## Author Contributions

**Conceptualization:** Hanaa S. Omar, Neama H. Osman, Mohamed A. Abou-Zeid.

**Data curation:** Hanaa S. Omar, Mohamed A. Abou-Zeid.

**Formal analysis:** Hanaa S. Omar, Abdullah Al Mutery, Nour El-Houda A. Reyad, Mohamed A. Abou-Zeid.

**Funding acquisition:** Hanaa S. Omar.

**Investigation:** Hanaa S. Omar, Mohamed A. Abou-Zeid.

**Methodology:** Hanaa S. Omar, Neama H. Osman.

**Project administration:** Hanaa S. Omar.

**Resources:** Hanaa S. Omar, Mohamed A. Abou-Zeid.

**Software:** Hanaa S. Omar, Abdullah Al Mutery, Neama H. Osman, Nour El-Houda A. Reyad.

**Supervision:** Hanaa S. Omar.

**Validation:** Hanaa S. Omar.

**Visualization:** Mohamed A. Abou-Zeid.

**Writing – original draft:** Hanaa S. Omar, Abdullah Al Mutery, Neama H. Osman, Nour El-Houda A. Reyad, Mohamed A. Abou-Zeid.

**Writing – review & editing:** Hanaa S. Omar, Abdullah Al Mutery, Neama H. Osman, Nour El-Houda A. Reyad, Mohamed A. Abou-Zeid.

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
