## [Decision Letter · Decision Letter 0]

23 Jun 2021

PONE-D-21-08742

Molecular identification of leaf, stem rust resistance in eighteen Egyptian wheat genotypes and the antifungal activity of chitosan-copper nanoparticles by molecular docking analysis

PLOS ONE

Dear Dr. omar,

Thank you for submitting your manuscript to PLOS ONE. After careful consideration, we feel that it has merit but does not fully meet PLOS ONE’s publication criteria as it currently stands. Therefore, we invite you to submit a revised version of the manuscript that addresses the points raised during the review process.

We look forward to receiving your revised manuscript.

Kind regards,

Himanshu Sharma

Academic Editor

PLOS ONE

Additional Editor Comments (if provided):

The manuscript entitled by Molecular identification of leaf, stem rust resistance in eighteen Egyptian wheat genotypes and the antifungal activity of chitosan-copper nanoparticles by molecular docking analysis by hanaa omar has been critically reviewed by the competent reviewers and based on their comments I will not be positive and authors have to revise the manuscript critically according to the queries. So that all the mistakes will be rectify and have sound message for the audience.

My special comments to authors are:

1. There are lots of mistake so read the manuscript critically and edit it.

2. Improve the Results, Discussion and rewrite the conclusion.

3. The authors have used very small no of phenotype merely 18 and the number is not sufficient for phenotype of rust resistance, Peoples used a panel 200 or 300 phenotype to score the data for at three different places for 3 years. The gene pool is very narrow and results may be biased. Is it possible for authors to increase the number of samples?

In the light of these comments revise the manuscript critically.

Journal Requirements:

5. Please amend the manuscript submission data (via Edit Submission) to include authors Neama H.Osman, Nour El-Houda A. Reyad , Mohamed A. Abou-Zeid. 

7. Please upload a new copy of Figure 1,3,5,6 and 8 as the detail is not clear. Please follow the link for more information: " ext-link-type="uri" xlink:type="simple">https://blogs.plos.org/plos/2019/06/looking-good-tips-for-creating-your-plos-figures-graphics/"
https://blogs.plos.org/plos/2019/06/looking-good-tips-for-creating-your-plos-figures-graphics/”. 

8. Please ensure that you refer to Figure 5 in your text as, if accepted, production will need this reference to link the reader to the figure.

9. We note that Figures 1,2,3,4,6,7 and 8 in your submission contain copyrighted images. All PLOS content is published under the Creative Commons Attribution License (CC BY 4.0), which means that the manuscript, images, and Supporting Information files will be freely available online, and any third party is permitted to access, download, copy, distribute, and use these materials in any way, even commercially, with proper attribution. For more information, see our copyright guidelines: http://journals.plos.org/plosone/s/licenses-and-copyright.

 a. You may seek permission from the original copyright holder of Figures 1,2,3,4,6,7 and 8 to publish the content specifically under the CC BY 4.0 license.

10. Please include captions for your Supporting Information files at the end of your manuscript, and update any in-text citations to match accordingly. Please see our Supporting Information guidelines for more information: http://journals.plos.org/plosone/s/supporting-information. 

11.We suggest you thoroughly copyedit your manuscript for language usage, spelling, and grammar. If you do not know anyone who can help you do this, you may wish to consider employing a professional scientific editing service.

A copy of your manuscript showing your changes by either highlighting them or using track changes (uploaded as a *supporting information* file) .

Reviewers' comments:

Reviewer's Responses to Questions

**Comments to the Author**

1. Is the manuscript technically sound, and do the data support the conclusions?

Reviewer #1: Yes

Reviewer #2: Yes

2. Has the statistical analysis been performed appropriately and rigorously? 

Reviewer #1: Yes

Reviewer #2: Yes

3. Have the authors made all data underlying the findings in their manuscript fully available?

Reviewer #1: Yes

Reviewer #2: Yes

4. Is the manuscript presented in an intelligible fashion and written in standard English?

Reviewer #1: Yes

Reviewer #2: No

5. Review Comments to the Author

Reviewer #1: Comments:

1. Wheat production is negatively affected by the attack of stem and rust diseases in Egypt and worldwide.

Wheat production is negatively affected by the attack of stem and leaf rust diseases in Egypt and worldwide.

2. In this respect, out of 20 scot and SRAP primers eleven and fourteen primers, showed polymorphism in 18 evaluated wheat genotypes, respectively.

3. Query: scot (SCoT), check in all manuscript

4. These markers have been listed in numerous references because they are powerful in determining the genetic diversity of various crops, such as wheat (Abou-Deif, et al., 2013 and El-Moneim, 2019).

Query: only these are markers to show up diversity what about other SSR and ILP markers if any provide references.

5. In this respect, out of 20 scot and SRAP primers eleven and fourteen primers, showed polymorphism in 18 evaluated wheat genotypes, respectively. SCoT and SRAP ones generated 140 and 121 polymorphic bands with 97 and 99% polymorphism, respectively

Query: rate of polymorphism is very high, is any positive or negative control were taken ?

6. Diverse wheat genotypes belong to same background or different?

Discussion section

7. Write concluding lines that showed the implementation of chitosan nanoparticles methods mostly suitable for which type of rust infection whether for leaf or stem.

8. Docking discussion is missing put concluding lines about what are the future aspect and what you conclude from this data

9. Heat map quality is poor improve it, check all image quality

10. What is the actual product size of eleven ScoT primers

11. What is the future aspect of overall study?

12. Insert line number in all manuscript

Reviewer #2: The manuscript is a useful work and may be considered after the following comments are addressed.

Title needs to be changed.

Abstract:

Uniform primer name SCoT not Scot.

It is unclear in abstract if the chitosan nanoparticles play important role in stem or leaf rust or both?

You mentioned SRAP and SCoT markers are useful for microbial diversity and cultivar? Why they are suitable in case of wheat? Why not other markers such as SSR or SNP?

Why diversity analysis was performed in wheat genotypes? What was its relation with rust?

How the authors avoided the rust pathogen DNA during molecular fingerprinting? These markers will amplify every DNA.

Give a brief description about each marker type used and their advantages.

CuChNp mention full form at its first occurrence in text.

Why SWISS model was preferred for modelling MAPK?

What are ideal criteria for a homology model to be stable and good for analysis?

Elaborate methods for structure modelling, validation (parameters), ligand-protein interaction and docking. Make it more clear.

Table 4 and 5 Where are high infection genotypes as mentioned in foot notes?

Using what data and how heat map was generated. Mention in the methods.

6. PLOS authors have the option to publish the peer review history of their article (what does this mean?). If published, this will include your full peer review and any attached files.

Reviewer #1: **Yes: **Pankaj Kumar

Reviewer #2: No

---

## [Author Response · Author response to Decision Letter 0]

24 Aug 2021

Authors reply list

To: Professor Vaughan Hurry

Manuscript Title: Molecular identification of leaf, stem rust resistance in eighteen Egyptian wheat genotypes and the antifungal activity of chitosan-copper nanoparticles by molecular docking analysis.

Corresponding author: Hanaa8324@yahoo.com

Hanaa S. Omar1,2 Abdullah Al Mutery ,3,4 , Neama H.Osman , 1| Nour El-Houda A. Reyad5, Mohamed A. Abou-Zeid6

Dear Professor 

We are grateful to the editors and reviewers for their time and constructive comments on our manuscript. We think that reviewer's constructive comments clearly contributed in improving the manuscript. We have carefully addressed all the comments as possible. The corresponding changes and refinements made in the revised paper are summarized below.

Line No. Existing to be written

 1. There are lots of mistake so read the manuscript critically and edit it. In response to this comment we edited it as required in all manuscript.

We tried our best to improve the English language by the help of professional grammar checker.

 2. Improve the Results, Discussion and rewrite the conclusion. It has already done with improvements to the Results, discussion rewrite the conclusion sections

All 3. The authors have used very small no of phenotype merely 18 and the number is not sufficient for phenotype of rust resistance, Peoples used a panel 200 or 300 phenotype to score the data for at three different places for 3 years. The gene pool is very narrow and results may be biased. Is it possible for authors to increase the number of samples? This study was carried out on Egyptian wheat genotypes. Besides, the numbers of Egyptian wheat genotypes are small.

 These are the Egyptian genotypes that have been selected from the breeding programs for the implementation of our current study. The aim is a complete study of the genetic diversity of these genotypes to know their sensitivity and resistance to stem and leaf rust disease, and then the application of nanotechnology used in resisting this disease. Moreover, the molecular docking analysis was also used to assess the Cu-chitosan nanoparticle treatment and its mode of action in disease control management. We can't increase the number of cultivars, because this requires waiting for two consecutive years, and therefore it is difficult to do so. 

Reviewer1

Dear reviewer1, we as authors appreciate your valuable and precious comments which considerably helped to improve and strengthen the manuscript. We are to provide the following replies hoping you will find them satisfactory and properly addressing your concerns. We have addressed the reviewer’s suggestions and revised the manuscript accordingly. Please find attached a detailed point by point response to the points of concern.

Comments from the editors and reviewers:

Line No. Existing to be written

 Wheat production is negatively affected by the attack of stem and rust diseases in Egypt and worldwide.

Wheat production is negatively affected by the attack of stem and leaf rust diseases in Egypt and worldwide. It has already done with removed that paragraph and added new paragraph to the abstract section.

Wheat has a remarkable importance among cereals worldwide. Wheat stem and leaf rust constitute the main threats that destructively influence grain quality and yield production. Pursuing resistant cultivars and developing new genotypes including resistance genes is believed to be the most effective tool to overcome these challenges.

 In this respect, out of 20 scot and SRAP primers eleven and fourteen primers, showed polymorphism in 18 evaluated wheat genotypes, respectively. In response to this comment we edited it the correct the paragraph in results section 

 Query: scot (SCoT), check in all manuscript In response to this comment we edited it the correct way in all manuscript 

 4. These markers have been listed in numerous references because they are powerful in determining the genetic diversity of various crops, such as wheat (Abou-Deif, et al., 2013 and El-Moneim, 2019). We already modify this part in discussion section 

In response to this comment, we edited it the correct way as required in this paragraph and in almost all results and discussion of SCOT and SRAP sections in the manuscript.

Abou-Deif, et al., 2013 used the ISSR molecular marker to estimate the genetic diversity between some Egyptian wheat genotypes. But El-Moneim, 2019 and Gowayed and El-Moneim (2021) used ISSR and SCoT primers to determine and evaluate some Egyptian wheat genotypes tolerant to drought and salt stress, respectively. But our current study used different primers for SCoT in genetic diversity and evaluation of Egyptian wheat genotypes' resistance to stem and leaf rust diseases.

5 Query: only these are markers to show up diversity what about other SSR and ILP markers if any provide references. In response to this comment we edited it as required in discussion section for SSR reference.

 Rate of polymorphism is very high, is any positive or negative control were taken? The rate of polymorphism is very high in positive control 

6 Diverse wheat genotypes belong to same background or different? Different background

Discussion section

7. Write concluding lines that showed the implementation of chitosan nanoparticles methods mostly suitable for which type of rust infection whether for leaf or stem. In response to this comment we edited in abstract, results, discussion and conclusion sections and added a new table (10) for leaf rust disease.

 Docking discussion is missing put concluding lines about what are the future aspect and what you conclude from this data In response to this comment, we edited it as required in discussion and conclusion sections.

9. Heat map quality is poor improve it, check all image quality We change and improve the figure of Heat map and added a new figure for PCA in our results and improved all figures.

10 What is the actual product size of eleven ScoT primers We can add the product sizes for SCoT and SRAP in tables (7 and 8) and result sections 

11. What is the future aspect of overall study? In response to this comment we edited it as required in conclusion section

12. Insert line number in all manuscript Was done 

Reviewer2

Dear reviewer, we thank you for the constructive remarks and comments on the manuscript. We have taken the comments on board to improve and clarify the manuscript. Please find below a detailed point-by-point response to all comments.

paragraph Title needs to be changed. We already change title manuscript to

Genetic Diversity, Antifungal Evaluation and Molecular Docking Studies of Cu-chitosan nanoparticles as Prospective Stem rust Inhibitor Candidates among some wheat genotypes 

 Uniform primer name SCoT not Scot. It was already done.

 It is unclear in abstract if the chitosan nanoparticles play important role in stem or leaf rust or both? It was already done

the chitosan nanoparticles play important role in both stem or leaf rust 

 You mentioned SRAP and SCoT markers are useful for microbial diversity and cultivar? Why they are suitable in case of wheat? Why not other markers such as SSR or SNP? In response to this comment we edited it as required in discussion. 

The gene-targeted molecular markers (SCoT) and sequence-related amplified polymorphisms (SRAP) analyses were used in the current study to determine their elevated power in polymorphism identification in wheat. The SCoT and SRAP markers were used to estimate and evaluate the genetic diversity among Egyptian wheat to stem and leaf rust disease. The results revealed that the gene-targeted molecular marker (SCoT) has several advantages over the use of dominant random markers (such as ISSR, RAPD, and AFLP). However, these markers reveal genetic diversity from the genic region in the genome and this functional diversity can be used in any species (Paliwal et.al., 2013).

Adil et.al., (2014) reported that the Simple Sequences Repeats (SSR) and SRAP were fast, accurate, and high throughput fingerprinting could be acquired using those markers, from the combined analysis, which revealed the existence of significant variation among the 33 accessions.

So in this study used SRAP as new technique to estimate genetic diversity between wheat genotypes .

 Why diversity analysis was performed in wheat genotypes? What was its relation with rust? In response to this comment we edited it as required in discussion. 

Wheat stem and leaf rust constitute the main threats that destructively influence grain quality and yield production.

Genetic diversity was used to detect leaf and stem rust-resistant wheat cultivars. 

And then the activity of chitosan nanoparticles (CuChNp) was applied on susceptible to stem and leaf rust in studied Egyptian wheat genotypes.

 How the authors avoided the rust pathogen DNA during molecular fingerprinting? These markers will amplify every DNA. DNA isolation from Egyptian wheat genotypes without infection to estimate genetic diversity. Pathogenic DNA doesn't exist in samples.

 Give a brief description about each marker type used and their advantages. In response to this comment we edited in discussion section

The gene-targeted molecular marker (SCoT) has several advantages over the use of dominant random markers (such as ISSR, RAPD, and AFLP). However, these markers reveal genetic diversity from the genic region in the genome and this functional diversity can be used in any species (Paliwal et.al.,2013). Moreover, the SRAP marker system is a new, simple, and efficient marker system that can be adapted for a variety of purposes, such as linkage map construction (Yeboah et al. 2007), genomic and cDNA fingerprinting gene tagging (Li and Quiros 2001), and genetic diversity analysis.

We was added this part in discussion section 

 CuChNp mention full form at its first occurrence in text. Was done

 Why SWISS model was preferred for modelling MAPK? SWISS-MODEL is powerful and valuable tool for homology modeling. There are many references that use the Swiss model in homology modeling , (Cuccioloni et al. 2017 ) , (Waterhouse, et al., 2018) and Cuccioloni et al. 2020

 What are ideal criteria for a homology model to be stable and good for analysis? SWISS-MODEL is powerful and valuable tool for homology modeling (Cuccioloni et al. 2020).

 Elaborate methods for structure modelling, validation (parameters), ligand-protein interaction and docking. In response to this comment we edited it as required in material and method section.

 Using what data and how heat map was generated. Mention in the methods. Make it more clear. In response to this comment we edited it as required in material and method section.

 Table 4 and 5 Where high infection genotypes are as mentioned in foot notes?

 Was done

---

## [Decision Letter · Decision Letter 1]

15 Sep 2021

Genetic Diversity, Antifungal Evaluation and Molecular Docking Studies of Cu-chitosan nanoparticles as Prospective Stem rust Inhibitor Candidates among some Egyptian wheat genotypes

PONE-D-21-08742R1

Dear Dr. Omar,

We’re pleased to inform you that your manuscript has been judged scientifically suitable for publication and will be formally accepted for publication once it meets all outstanding technical requirements.

Kind regards,

Himanshu Sharma

Academic Editor

PLOS ONE

Additional Editor Comments (optional):

Based on the recommendations of the reviewers manuscript is accepted for publication, there are always possibilities for improvement So authors can improve the manuscript at the time of proofread.

Reviewers' comments:

Reviewer's Responses to Questions

**Comments to the Author**

1. If the authors have adequately addressed your comments raised in a previous round of review and you feel that this manuscript is now acceptable for publication, you may indicate that here to bypass the “Comments to the Author” section, enter your conflict of interest statement in the “Confidential to Editor” section, and submit your "Accept" recommendation.

Reviewer #1: All comments have been addressed

Reviewer #2: All comments have been addressed

2. Is the manuscript technically sound, and do the data support the conclusions?

Reviewer #1: Yes

Reviewer #2: Yes

3. Has the statistical analysis been performed appropriately and rigorously? 

Reviewer #1: Yes

Reviewer #2: Yes

4. Have the authors made all data underlying the findings in their manuscript fully available?

Reviewer #1: Yes

Reviewer #2: Yes

5. Is the manuscript presented in an intelligible fashion and written in standard English?

Reviewer #1: Yes

Reviewer #2: Yes

6. Review Comments to the Author

Reviewer #1: Authors fulfill all comments now, i think its more stream line and ready for publishing.

Reviewer #2: The questions raised were appropriately answered. The revised manuscript may be considered for publishing.

7. PLOS authors have the option to publish the peer review history of their article (what does this mean?). If published, this will include your full peer review and any attached files.

Reviewer #1: No

Reviewer #2: No

---

## [Editor Report · Acceptance letter]

19 Oct 2021

PONE-D-21-08742R1 

Genetic Diversity, Antifungal Evaluation and Molecular Docking Studies of Cu-chitosan nanoparticles as Prospective Stem rust Inhibitor Candidates among some Egyptian wheat genotypes 

Dear Dr. Omar:

I'm pleased to inform you that your manuscript has been deemed suitable for publication in PLOS ONE. Congratulations! Your manuscript is now with our production department. 

Kind regards, 

on behalf of

Dr. Himanshu Sharma 

Academic Editor

PLOS ONE